# HS-SFT: Hybrid Sparse Supervised Fine-tuning for Offline LLM KV Cache Eviction

## Abstract

Long-context LLMs are constrained by the linear growth of key–value (KV) caches during autoregressive decoding, which incurs pronounced latency and memory overhead. KV eviction mitigates this issue, with existing efforts fall into offline policies with fixed eviction patterns and online policies that adaptively discard cache based on attention scores. While online eviction typically preserves accuracy under standard benchmarks, its performance can collapse in practical multi-turn dialogue scenarios where the query positions vary, and integration with pre-fill acceleration remains challenging. In contrast, offline eviction is infrastructure-friendly and generalizable but commonly sacrifices more accuracy. In this paper, we explore Supervised Fine-Tuning (SFT) for offline KV eviction and demonstrate its efficacy as a simple and powerful alternative to the design of complex online eviction metrics. We further propose Hybrid Sparse Supervised Fine-Tuning (HS-SFT) to explore the optimal offline design of KV eviction within SFT. In particular, HS-SFT employs a straight-through estimator to learn discrete local-window allocations of streaming heads across layers with budget-aware balancing loss, such that under high compression ratios—where dense-head capacity is constrained—the budget can be more effectively skewed to capture critical information. Across extensive evaluations on a wide array of LLMs and long-context tasks, HS-SFT delivers substantial performance gains over state-of-the-art eviction baselines, only consuming fewer than 4 hours of SFT on a single 8-GPU node. These results position training-aware offline eviction—achieved with simple SFT—as an effective and practical path to scalable long-context inference. Code will be available.

## 1 Introduction

Long-context large language models (Liu et al., 2024b; Google et al., 2024) have demonstrated strong capabilities across real-world applications, including multi-turn dialogue (Li et al., 2025; Taori et al., 2023), long-document summarization (Goyal & Durrett, 2020; Zhang et al., 2024), in-depth reasoning (Guo et al., 2025; Zelikman et al., 2022), and repository-level code generation (Zhang et al., 2023a; Zhang et al.). Nevertheless, the attention mechanism (Vaswani et al., 2017) underpinning state-of-the-art LLMs incurs substantial computational and memory overheads in long-context scenarios. Specifically, autoregressive decoding requires caching the keys and values of all preceding tokens to preclude repetitive computation, yielding a key–value (KV) cache whose size scales linearly with input length. This linear scaling substantially inflates peak memory footprint, exacerbates end-to-end latency, and impedes large-scale deployment. For instance, the Llama-3-70B-Instruct model (Dubey et al., 2024), when run with a batch size of 32 and a 128k-context window, consumes more than 1 TB of KV-cache footprint at FP16 precision—rendering LLM serving prohibitively expensive.

A multitude of efforts have been undertaken to surmount this inference challenge, with major methods falling broadly into two paradigms: KV selection and KV eviction. The former group dynamically loads KV cache of the most relevant blocks/tokens to the current query to reduce computation cost with all caches stored in memory. Notable methods such as Quest (Tang et al., 2024b) and NSA (Yuan et al., 2025) have shown remarkable capacity to enhance LLM throughput on long sequences, while closely matching the performance of full attention. Nonetheless, KV selection does not evict any cache from memory and therefore results in no KV footprint reduction. This paper hereby centers on KV eviction (Xiao et al., 2023; Zhang et al., 2023b; Xiao et al., 2025), which directly removes less important entries from the KV cache, thereby improving both decoding and footprint efficiency.

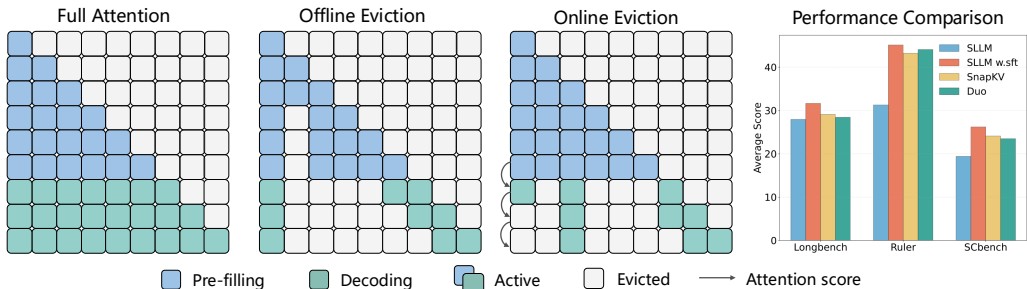

Figure 1: (left) Toy illustration of offline and online KV-cache eviction methods. Offline eviction predefines a fixed eviction pattern prior to inference and is therefore compatible with prefill acceleration. In contrast, online eviction dynamically evicts tokens during prefill based on historical attention scores. (right) SFT effectively elevates the performance of offline eviction method Stream-SLLM) (Xiao et al., 2023) to that of the online eviction method SnapKV (Li et al., 2024).

We heuristically categorize KV eviction methods into *offline* and *online* paradigms, based on whether the eviction policy is predetermined before inference. A canonical example of offline eviction is StreamingLLM (Xiao et al., 2023), which predefines sink and local attention window as a fixed sparsity pattern. Offline eviction offers a simple yet efficient solution that can seamlessly be integrated with off-the-shelf AI infrastructures (Ye et al., 2025; Dao et al., 2022). Conversely, online eviction employs heuristic metrics to dynamically devise eviction policies, mostly drawing upon attention scores during pre-filling. For instance, SnapKV (Li et al., 2024) evicts KV cache based on aggregated attention scores between the input context and the observation window at the prompt end. While online eviction generally achieves superior performance compared to offline eviction on common tasks, notable weaknesses also exist. First, in practical settings such as multi-turn dialogue or other cases where queries do not occur at the prompt end, online eviction can suffer from severe performance degradation (Xiao et al., 2025). Second, efficiency concerns arise from the incompatibility with pre-filling acceleration (Bhaskar et al., 2025). Therefore, a pivotal problem emerges: how to unify the strengths of task generalization, decoding efficiency from offline eviction and long-context performance retention from online eviction?

In this paper, we demonstrate that supervised fine-tuning (SFT) could be a simple yet powerful avenue to address this challenge. Figure 2 reveals that: (1) SFT substantially narrows the performance disparity of StreamingLLM in comparison to the online eviction method SnapKV; (2) the performance gap among different online eviction methods is significantly minimized following SFT. This uncovers a motivating conclusion that LLMs can inherently learn to alleviate the performance degradation induced by cache eviction during SFT. This insight also resonates with NSA (Yuan et al., 2025) and GPT-OSS (OpenAI, 2025), which incorporate KV/attention sparsity during the pre-training phase and show promising native sparse performance. Differently, our findings suggest a more efficient way that directly applying lightweight SFT to off-the-shelf LLMs can also effectively rehabilitate the performance of online eviction paradigm.

We go further to investigate the optimal fine-tuning paradigm for offline KV eviction. Current eviction methods predominantly concentrate on the allocation ratio between dense and streaming heads. Here, the streaming heads are set with a fixed 128/256 local window size, which, however, imposes a significant constraint on performance preservation under high sparsity regimes. For example, under a 2048-token budget, Duo-attn can designate only 1% of heads as dense, confining the rest to local windows of 256 tokens. To address this shortcoming, we propose Hybrid Sparse Supervised Fine-Tuning (HS-SFT) for offline KV eviction. In particular, HS-SFT jointly learns optimal local window size allocations across layers, thereby enabling more strategic cache budget distribution under high sparsity to optimize performance retention. To realize this, we utilize a Straight-Through Estimator (STE) (Bengio et al., 2013) to learn layer-wise budget allocation policies, supplemented by a budget-aware balance loss that explicitly promotes KV sparsity.

Experiments on a wide variety of models and downstream tasks demonstrate that HS-SFT achieves superior trade-offs among accuracy, latency, and memory footprint reduction for KV cache eviction. For instance, by applying HS-SFT to fine-tune the LLaMA-3-8B-1048K model (Dubey et al., 2024)

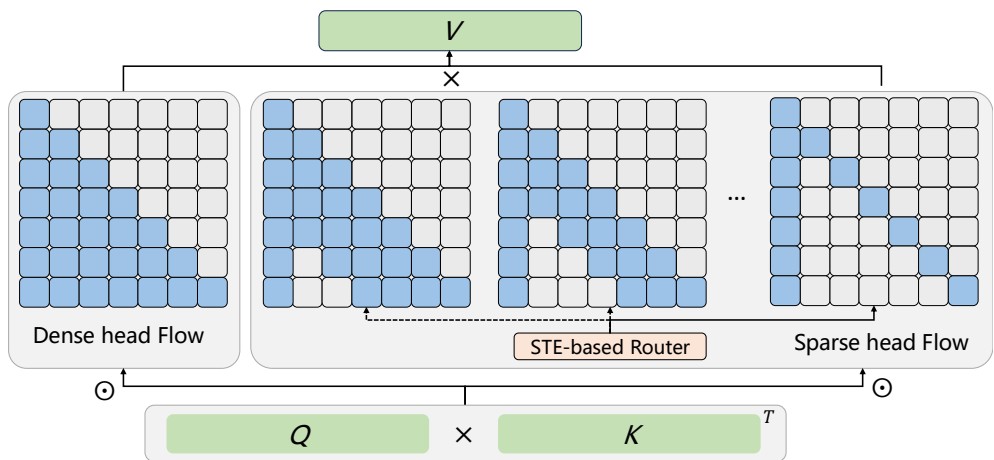

Figure 2: Framework of HS-SFT. We use two types of attention masks in each layer during training. The dense head flow performs full attention computations, while the sparse head flow learns optimal local window budgets through layer-specific STE-based router.

in less than 4 hours on a single 8-A800 node, we achieve a $2.8\times$ end-to-end speedup and a $1.8\times$ reduction in memory usage at 100K decoding length with 10% KV budget, while surpassing Duo-attn by 23.7% to 29.5% average score on the LongBench (Bai et al., 2023) benchmark. We hope this work lays a foundation for future innovations focused on training-aware KV eviction strategies.

## 2 HS-SFT

### 2.1 Exploring SFT for Offline KV Eviction

We explore offline KV cache eviction for efficient long-context LLM inference. In the offline setting, the eviction plan is fixed prior to inference and typically entails replacing the vanilla dense attention heads (Vaswani et al., 2017) with streaming heads (Xiao et al., 2023). Concretely, a streaming head retains a small set of sink tokens at the sequence prefix together with a fixed-size local window that slides throughout decoding. Such offline definition for KV sparse pattern confers practical deployment advantages (e.g., compatibility with prefilling acceleration (Xiao et al., 2025)). However, relative to online methods that dynamically determine eviction policies using attention scores (Zhang et al., 2023b; Li et al., 2024), offline schemes often suffer pronounced performance degradation.

In this work, we investigate supervised fine-tuning (SFT) (Raffel et al., 2020; Ouyang et al., 2022) to recover the performance of offline KV eviction without compromising its offline nature. Specifically, we conduct a lightweight SFT on 1B tokens from UltraChat (Ding et al., 2023) under the predefined sparse pattern of a representative eviction method StreamingLLM (Xiao et al., 2023). Remarkably, SFT enables the pre-trained LLM to mitigate the degradation induced by offline KV eviction, yielding substantial improvements across diverse long-context scenarios, as shown in Figure 2. SFT also markedly narrows the performance disparity among different offline eviction schemes, suggesting that—rather than devising intricate metrics—model fine-tuning offers an effective and compelling alternative for improving KV-eviction performance. Our findings also align with GPT-OSS (OpenAI, 2025), which replaces a subset of layers with streaming layers during pre-training and demonstrates robust performance of such offline eviction paradigm. Nevertheless, we posit that even modest SFT suffices to endow the model with global information aggregation capabilities under KV cache eviction, offering a practical and efficient solution.

### 2.2 Hybrid Sparse SFT

We further explore the optimal SFT paradigm for offline KV-cache eviction. Existing approaches predominantly optimize hybrid allocations between dense and streaming heads (Bhaskar et al., 2025; Xiao et al., 2025; OpenAI, 2025), where streaming heads operate with small, fixed local windows

(e.g., 128/512 tokens). The motivation stems from that attention heads of transformer-based LLM exhibit distinct and stable specialization patterns (Wu et al., 2024; Xiao et al., 2025): retrieval heads capture global information, whereas streaming heads prioritize recent tokens and attention sinks.

However, such hard dense–streaming hybrid management faces two limitations under high KV sparsity. First, while strongly distinct head behaviors are visible in standard multi-head attention (MHA) (Vaswani et al., 2017), they are less pronounced in grouped-query attention (GQA) (Ainslie et al., 2023) with compact head dimension design (Xiao et al., 2025). Since most modern models adopt GQA, a fixed dense–streaming split becomes inflexible at high sparsity. Second, under extremely high eviction rates like 90%, only a small subset of heads remain dense, whereas other globally oriented heads are forced into overly tight streaming windows and thus suffer pronounced performance degradation.

To address these issues, we propose *Hybrid Sparse Supervised Fine-Tuning* (HS-SFT) for offline KV eviction. The key principle of HS-SFT falls in that it learns soft local window budget candidates during SFT, in conjecture with dense head, enabling more strategic cache-budget allocation under high sparsity to maximize performance retention. Concretely, during training, we keep a fixed fraction $\alpha$ of heads per layer dense, and learn the optimal local window budget allocations for the remaining heads. Here each layer is equipped with a discrete budget set $\mathcal{B} = [b_1, b_2, \ldots, b_m]$ At forward passes, each sparse head selects at most $k$ tokens from the current KV cache to attend to, where $k \in \mathcal{B}$. This reduces both attention FLOPs and memory footprint and, crucially, allows layer-wise specialization of KV eviction based on different patterns of streaming heads.

**STE-based Learnable Budget Selection.** For the sparse head flow, we learn the optimal budget $k \in \mathcal{B}$. We associate each candidate budget $b_i$ with a learnable logit $z_i$, and let $\mathbf{z} = [z_1, \ldots, z_m]$ denote the logits for streaming head in one specific layer*. We perform hard selection in the forward pass as

$$k = \mathcal{B}\left[\arg\max_i z_i\right]. \tag{1}$$

Since $\arg\max$ is non-differentiable, we adopt the Straight-Through Estimator (STE) (Bengio et al., 2013) to optimize $\mathbf{z}$. Particularly, we pass the gradient only to the selected logit and zero out the rest:

$$\frac{\partial \mathcal{L}_{\text{LM}}}{\partial z_i} = \begin{cases} \frac{\partial \mathcal{L}_{\text{LM}}}{\partial \text{output}} & \text{if } i = \arg\max_j z_j, \\ 0 & \text{otherwise.} \end{cases} \tag{2}$$

This preserves hard budget selection during forward propagation while enabling end-to-end optimization for the budget logits of each layer.

**Budget-aware Balance Loss.** To avoid trivial solutions where all sparse heads choose large budgets, we regularize the discrete choice via a KL divergence to a prior that favors smaller budgets. Let $m = |\mathcal{B}|$, we derive the selection distribution as

$$p_i = \text{softmax}(\mathbf{z})_i, \quad i \in \{1, \ldots, m\}. \tag{3}$$

Then, we define a prior over indices that increases mass on smaller budgets via a power law:

$$q_i = \frac{(m + 1 - i)^\gamma}{\sum_{j=1}^{m}(m + 1 - j)^\gamma}, \quad \gamma > 0. \tag{4}$$

The budget-aware balance loss is then derived by the KL divergence $\text{KL}(p\|q)$:

$$\mathcal{L}_{\text{balance}} = \lambda \sum_{i=1}^{m} p_i \big( \log(p_i + \varepsilon) - \log(q_i + \varepsilon) \big), \tag{5}$$

where $\lambda > 0$ controls the sparsity penalty and $\varepsilon > 0$ is a small constant for numerical stability. The overall objective is

$$\mathcal{L} = \mathcal{L}_{\text{LM}} + \mathcal{L}_{\text{balance}}. \tag{6}$$

This objective explicitly encourages compact budgets whenever possible, while permitting larger budgets where necessary to preserve performance.

---

*We refrain from assigning learnable budget logits to every attention head within a layer, as it induces numerous heterogeneous KV sparse inference flows that markedly degrades inference efficiency.

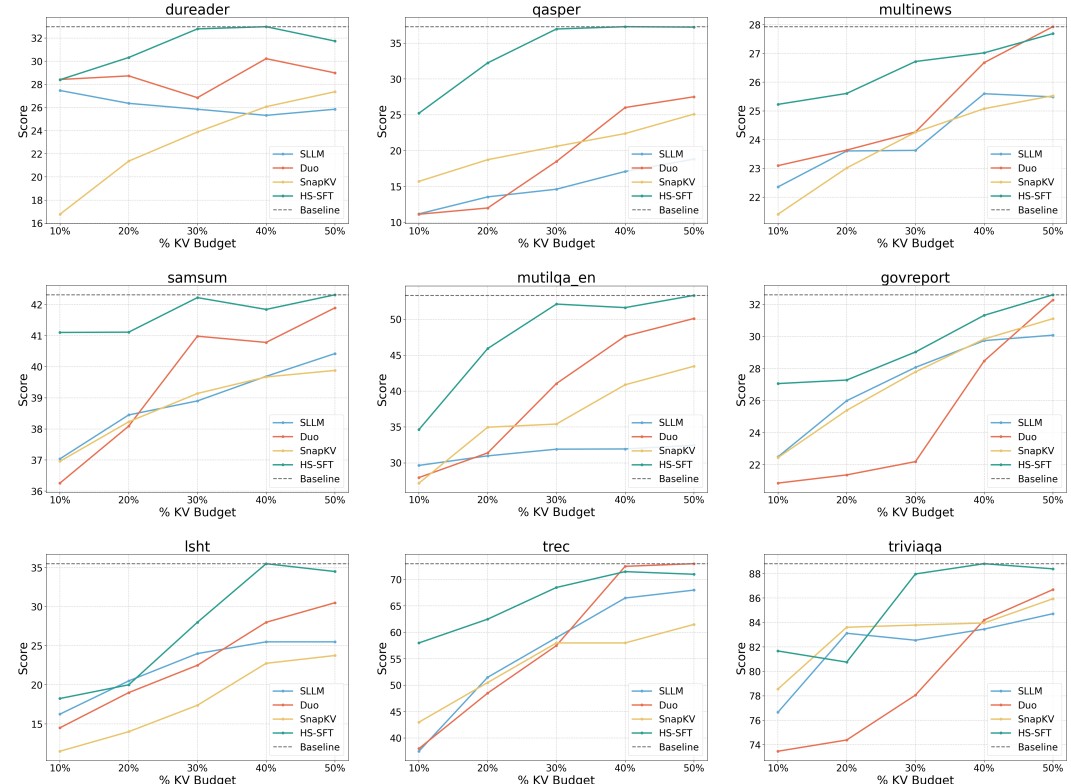

Figure 3: Per-task LongBench results on Llama-3-8B-Instruct-1048K. HS-SFT consistently narrows the gap to dense attention across KV budget percentage and remains stable across task types.

## 2.3 HS-SFT WORKFLOW

The training and inference pipeline of HS-SFT is illustrated in Figure 2. Prior to training, we initialize a fraction $\alpha$ of dense heads using the logit map optimized by Duo-Attn (Xiao et al., 2025). During training, we jointly optimize the layer-wise local-window budget logits and the model weights. Given a target sparsity $\rho$ at the inference stage, we retain the dense heads (fraction $\alpha$) and allocate the remaining cache to streaming heads by monotonically scaling the learned budgets of each layer to meet the overall sparsity constraint. A detailed algorithm workflow is presented in Appendix A.3.

HS-SFT unifies the strengths of offline eviction with the performance retention typically attributed to online heuristics. By hard-selecting discrete budgets via STE and regularizing the selection distribution with a budget-aware loss, HS-SFT learns optimal layer-wise sparsity patterns for streaming heads while jointly adapting the pre-trained LLM to mitigate performance degradation under KV eviction. At inference time, the execution remains strictly offline: a subset of heads stays dense, and all remaining heads run streaming attention with their layer-wise calibrated budgets derived from the learned preferences. As a result, HS-SFT maintains the efficiency of offline eviction paradigms, while effectively mitigating the performance gap versus dense attention.

## 3 EXPERIMENTS

### 3.1 EXPERIMENTAL SETTINGS

**Tasks, Models, and Baselines.** We assess long-context capabilities across three representative benchmarks: LongBench (Bai et al., 2023), Ruler-16K (Hsieh et al., 2024), and Needle-in-a-Haystack (NIAH) (Kamradt, 2024). We evaluate two base long-context LLMs instantiated in three context-window configurations: Llama-2-7B with 32K extension Touvron et al. (2023) and Llama-3-8B-Instruct with Gradient-262K/1048K extensions (Pekelis et al., 2024). We compare offline and

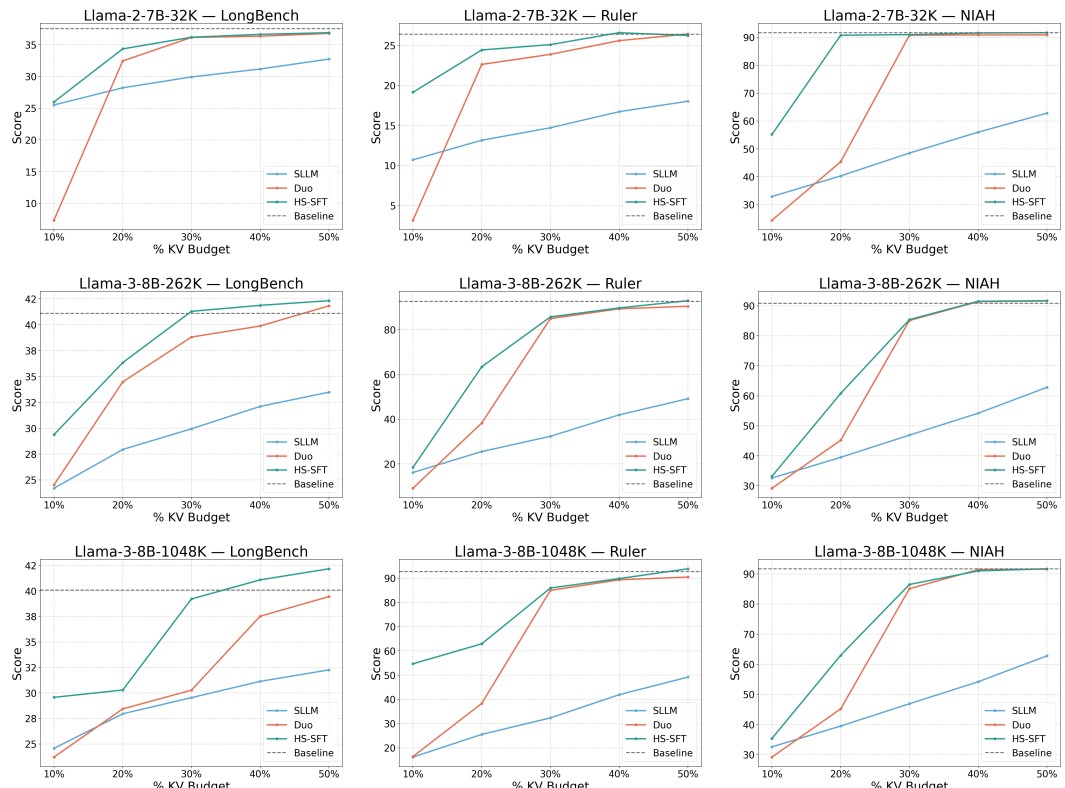

Figure 4: Average accuracy across Llama-2-7B-32K and Llama-3-8B-262K/1048K. The superiority of HS-SFT remains stable across different tasks and KV budgets.

online key–value (KV) cache eviction policies, including StreamingLLM (Xiao et al., 2023), Duo-Attention (Xiao et al., 2025), and SnapKV (Li et al., 2024). Consistent with prior work (Tang et al., 2024b; Xiao et al., 2025), we simulate generation of the final 50 tokens for elongated inputs during evaluation to ensure a fair comparison between offline and online regimes.

**Implementation Details.** Following Gao et al. (2024), we conduct supervised fine-tuning (SFT) on 1B tokens sampled from UltraChat-200K (Ding et al., 2023). Detailed SFT hyperparameters are provided in Appendix A.4. During training, we employ the block-sparse kernel for streaming attention as implemented by Guo et al. (2024), aligning configurations with Xiao et al. (2025). For budget learning within HS-SFT, we use a learning rate of $3 \times 10^{-4}$ with no weight decay. The candidate budget set is $\{1, 2, 3, 4, 5, 6, 7, 8\}$ blocks, where each block contains 128 tokens. To assess performance across KV budgets, we set the dense-head ratio $\alpha$ to be 5% lower than the targeted KV budget. Although a uniform dense-head ratio of $\alpha = 5\%$ can be monotonically rescaled during inference to satisfy a target sparsity budget, we observe that budget-specific $\alpha$ values yield more consistent performance gains.

## 3.2 QUANTITATIVE RESULTS

**LongBench.** LongBench comprises a diverse suite of long-context tasks, including retrieval-style QA, long-document summarization, and multi-passage classification that stress faithful cross-span integration and robust global-to-local information routing. Figure 3 presents a comparison of HS-SFT against competing methods across six representative LongBench tasks. More comprehensive results for all tasks are provided in Appendix A.9. Relative to existing offline eviction baselines, HS-SFT markedly reduces the performance gap to dense attention across KV eviction levels ranging from 50% down to 10%. Notably, the gains delivered by HS-SFT intensify as the KV budget shrinks, since learnable, head-wise budgets adaptively allocate capacity to layers where long-range aggregation is most advantageous.

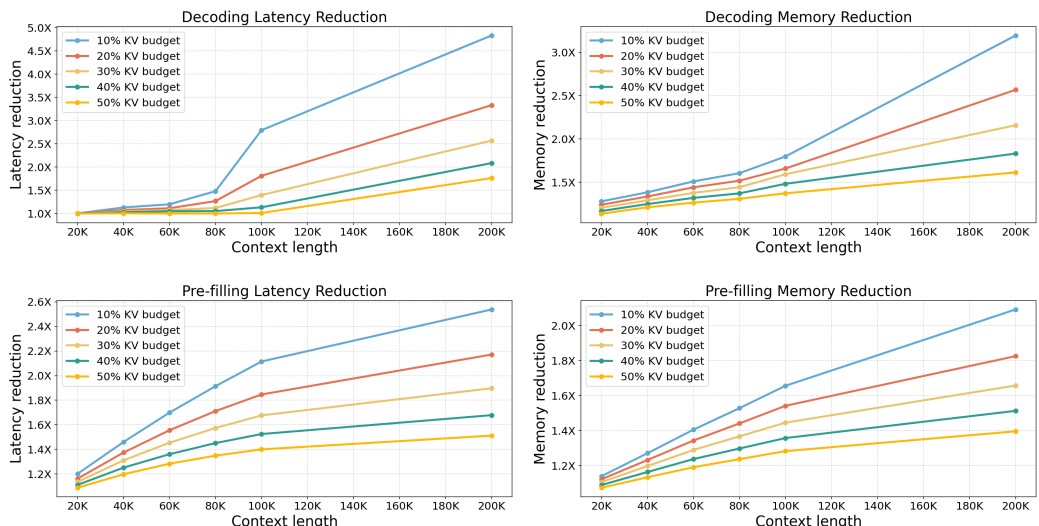

Figure 5: End-to-end efficiency results on one GPU (batch size=1).

Table 1: Performance comparison under identical SFT settings at 10% KV budget for Llama-3-8B-1048K. HS-SFT achieves the strongest average performance.

| KV budget | Duo-Attn | | SLLM | | LS | | HS-SFT |
| --- | --- | --- | --- | --- | --- | --- | --- |
| | Base | SFT | Base | SFT | Base | SFT | (Ours) |
| 50% | 39.4 | 40.1 | 32.2 | 36.8 | 30.8 | 37.1 | **42.2** |
| 10% | 24.1 | 26.3 | 23.8 | 27.1 | 23.1 | 26.4 | **29.6** |

**Cross-Model and Cross-Task Generalization.** We further evaluate Llama-2-7B-Instruct-32K, Llama-3-8B-Instruct-262K and Llama-3-8B-Instruct-1048K-Instruct on LongBench, Ruler, and NIAH. Ruler emphasizes long-range cross-document retrieval and compositional reasoning, whereas NIAH tests the efficacy of models to accurately retrieve relevant information from long context. As shown in Figure 4, across all evaluated LLMs and task suites, HS-SFT consistently outperforms existing baselines and matches or surpasses online methods at every KV budget. These consistent gains indicate strong transferability across pretraining distributions and task typologies.

**Comparison under Identical SFT Settings.** We conduct an ablation of offline eviction schemes under identical SFT configurations (data, steps, and hyperparameters) to isolate the effect of SFT. Specifically, Duo-Attention, StreamingLLM, and HS-SFT are fine-tuned on the same 1B-token UltraChat corpus and evaluated on all benchmarks at matched KV budgets. We additionally evaluate the hybrid strategy of GPT-OSS (OpenAI, 2025), which employs interleaved streaming layers (Layer Streaming, LS). As shown in Table 1, HS-SFT attains the highest average performance across datasets and sparsity levels. This suggests that learning discrete, layer-specific budget preferences via STE-based selection coupled with budget-aware regularization outperforms fixed-window approaches and rigid dense/streaming partitions. Moreover, HS-SFT delivers significant improvements over the GPT-OSS-style hybrid paradigm, underscoring its potential to scale to larger architectures and more demanding training regimes as a promising future work.

**Efficiency Analysis.** We measure end-to-end memory footprint and latency for the prefill and decode phases on a single NVIDIA GPU with a 4K chunk size. As shown in Figure 5, consistent with offline sparse methods (Xiao et al., 2025; 2023), HS-SFT substantially reduces both prefill latency and memory consumption. Decoding efficiency scales approximately linearly with KV-budget reduction, achieving a 2.8× speedup at a 100K-token context length under 10% KV budget. Coupled with its accuracy gains, HS-SFT lies on a favorable Pareto frontier within the offline eviction paradigm, improving quality while preserving deployment simplicity. We further analyze the inference efficiency of HS-SFT compared with traditional KV eviction (Li et al., 2024; Xiao et al., 2023), layer-wise hybrid (OpenAI, 2025) and head-wise hybrid (Fu et al., 2024b; Ge et al., 2023) paradigms in Appendix A.6.

Table 2: Ablations on SFT configurations. Updating dense heads yields significant gains, and SFT outperforms continued pretraining.

| Configuration | LongBench | Ruler |
|---|---|---|
| Frozen | 27.03 | 49.14 |
| Updated | **29.57** | **54.68** |
| CP | 24.55 | 27.07 |
| SFT | **29.57** | **54.68** |

Table 3: Regularizer ablation at 10% KV budget. Budget-aware KL regularization ($\lambda = 10^{-2}$) achieves optimal performance.

| Regularizer | $\lambda$ | LongBench | Ruler |
|---|---|---|---|
| L2 penalty | $10^{-2}$ | 25.51 | 39.51 |
| KL divergence | $10^{-3}$ | 27.10 | 41.87 |
| KL divergence | $10^{-2}$ | **29.57** | **54.68** |
| KL divergence | $10^{-1}$ | 28.44 | 50.12 |

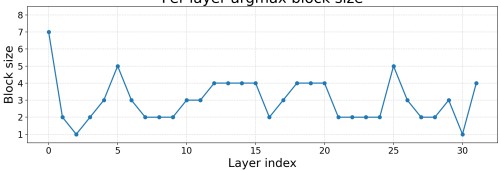

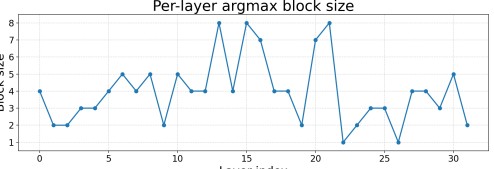

Figure 6: Learned budget allocation (blocks) across transformer layers of Llama-2-7B-32K and Llama-3-8B-1048K. Layer indices increase from left to right.

Table 4: Ablations on SFT scale and domain. The average score is computed on LongBench under 10% KV budget and with @ marking per-dataset mixing ratios when forming the SFT corpus.

| Dataset | Training Tokens | Avg. Score |
|---|---|---|
| *SFT size* | | |
| UltraChat@1.0 | 1B | 29.57 |
| UltraChat@1.0 | 0.5B | 28.76 |
| UltraChat@1.0 | 2B | 29.42 |
| *SFT Domain* | | |
| UltraChat@0.8 + Tulu-V3-Code@0.2 | 1B | 28.34 |
| UltraChat@0.8 + Tulu-V3-Math@0.2 | 1B | 28.48 |
| Tulu-V3@1.0 | 1B | 29.99 |
| UltraChat@0.8 + ArXiv-Sum@0.2 | 1B | 29.08 |
| ArXiv-Sum@1.0 | 1B | 26.12 |

## 3.3 ABLATION STUDIES

We analyze HS-SFT along two axes—budget learning and SFT configuration. All ablations use Llama-3-8B-Instruct-1048K with 10% KV budget.

**Budget Learning Mechanism.** We compare our budget-aware KL regularizer (Section 2.2) against a conventional L2 penalty, sweeping $\lambda \in \{10^{-3}, 10^{-2}, 5 \times 10^{-2}, 10^{-1}\}$. As shown in Table 3, $\lambda = 10^{-2}$ yields the best macro-average performance across benchmarks. Lower $\lambda$ values insufficiently penalize excessive budget allocation, whereas higher values over-suppress necessary long-range aggregation. Expanding the candidate budget set to $[1, 32]$ or $[8, 16]$ blocks did not improve performance, likely because more blocks yield local window sizes larger than the 1.5K average sequence length in the Ultrachat corpus, limiting the training efficacy for KV eviction paradigm. Figure 6 visualizes the learned layer-wise budgets, revealing unique distributions across different layers.

**SFT Configuration Analysis.** Table 2 examines SFT design choices. Freezing dense heads during fine-tuning consistently underperforms full-parameter updates, suggesting that adapting dense heads strengthens global retrieval. We also compare against continued pretraining (CP) on RedPajama (Weber et al., 2024) using an equivalent training tokens. As shown in Table 2, SFT outperforms CP, as token-level continued pretraining does not adequately instill global-information recovery under constrained KV budgets.

We further study how the SFT data domain and scale influence the performance of HS-SFT. Table 2 list the results. In particular, we first vary the number of training tokens on UltraChat to measure the size effect and observe a clear scaling trend on downstream tasks, yet continuing to increase the training tokens does not linearly lead to performance improvement. Next, we replace UltraChat with Tulu-V3 (Lambert et al., 2024) and also mix UltraChat with the Code and Math subsets of Tulu-V3 to examine domain sensitivity. Finally, we include data settings biased for summarization tasks using ArXiv-Summarization (Cohan et al., 2018) that highly correlate with long-context ability. While domain-specialized data can improve in-domain metrics, aggressively skewing toward summarization tends to reduce overall performance on general tasks.

**Dense Head Initialization.** We investigate the impact of dense-head initialization on HS-SFT. We consider two variants: (i) random initialization and (ii) selecting dense heads following RazorAttention (Tang et al., 2024a). As shown in Table 5, random selection of dense head incurs a notable performance drop compared with a strategic choice. Moreover, RazorAttention achieves performance close to Duo-Attention, suggesting that HS-SFT is relatively robust to the choice of initialization.

Table 5: Ablations on dense head initialization. The average score is computed on Long-Bench under a fixed 10% KV budget.

| Method | Avg. Score |
|---|---|
| Random | 26.41 |
| RazorAttention | 29.02 |
| Duo-Attention | **29.57** |

## 4 RELATED WORK

A diverse array of methodologies has been introduced to optimize the efficiency of KV cache in LLMs for long-context inference. These methods can be broadly grouped into the following categories.

### 4.1 CACHE EVICTION

Cache eviction refers to the removal of redundant KV entries to yield a more lightweight KV cache, thereby enhancing decoding efficiency and reducing memory footprint relative to conventional full-cache management paradigm. In this paper, we heuristically divide KV cache eviction methods into *offline* and *online* paradigms, depending on whether the eviction policy is fixed prior to inference.

**Offline KV Eviction.** Offline KV eviction enforces a static, sparse KV pattern irrespective of the input sequence, thereby ensuring consistent decoding latency and memory efficiency throughout inference. Representative examples include StreamingLLM Xiao et al. (2023), which persistently retains both the initial and most recent tokens within the sequence, and Duo-attn Xiao et al. (2025), which further optimizes dense and streaming head assignments to balance fidelity and efficiency under a pre-defined KV budget. The recently introduced GPT-OSS OpenAI (2025) also exemplifies an offline KV eviction paradigm, incorporating interleaved streaming layers during pretraining to natively support KV eviction. Offline eviction methods are distinguished by their seamless integration with existing inference frameworks, thereby delivering stable end-to-end throughput and memory consumption benefits; however, they often incur performance degradation under high sparsity regimes. Our proposed DS-SFT addresses these limitations from two perspectives while preserving the deployment advantages of offline eviction. First, we pioneer the exploration of SFT strategies tailored for KV eviction and demonstrate their efficacy in enhancing offline eviction performance. Second, DS-SFT enables more strategic local window size allocation to optimize better performance than existing methods that keep unified local window size across all layers.

**Online KV Eviction.** Online KV eviction dynamically modulates the retention and removal of KV entries during inference, guided by preceding attention scores Zhang et al. (2023b); Li et al. (2024); Fu et al. (2024a); Cai et al. (2024). For instance, H2O Zhang et al. (2023b) leverages accumulated attention scores as a criterion to selectively preserve salient KV entries. SnapKV Li et al. (2024) further exploits local window attention scores over the prefilling context to evict token caches with minimal contribution. Building upon SnapKV, PyramidKV Cai et al. (2024) heuristically allocates more cache to lower layers while reducing allocation in higher layers. MorphKV Ghadia et al. (2025) further refines the compressed KV cache via lightweight updates guided by attention patterns of recent tokens. Online eviction generally achieves superior performance than offline eviction on standard long-context tasks. Nevertheless, the heuristic measurement based on attention scores may

precipitate significant performance degradation in practical multi-turn dialogue scenarios, particularly when queries are not situated at the end of the prompt Xiao et al. (2025). Moreover, online eviction strategies are inherently at odds with prefill-acceleration techniques Bhaskar et al. (2025), further constraining their efficiency in handling tasks characterized by extensive pre-filling, such as document summarization. In this work, we focus on the context of KV eviction to strike better generalization and deployment efficiency.

A line of recent works has also investigated head-wise KV compression (Fu et al., 2024b; Ge et al., 2023). Fastgen Ge et al. (2023) pioneered head-wise KV eviction by eliminating long-range contexts in heads that emphasize local patterns, discarding non-special tokens in heads focused on special tokens, and retaining the standard KV cache only for heads with broad attention coverage. HeadKV Fu et al. (2024b) further proposed an importance-score estimation method that jointly evaluates each head's retrieval and reasoning capabilities. While HS-SFT also involves partitioning along the head dimension, we diverge by classifying heads into dense and sparse flows on a per-layer basis, as shown in Figure 2. This design guarantees exceptionally high inference efficiency, whereas fine-grained head-wise implementations typically require specialized kernels and are difficult to apply effectively in training scenarios. Therefore, we refrain from assigning learnable budget logits to every attention head within a layer, as it induces numerous heterogeneous KV sparse inference flows that markedly degrades training/inference efficiency. We provide a detailed efficiency analysis in Appendix A.6.

## 4.2 Cache Selection

Cache selection methods preserve the entirety of KV entries in memory, yet dynamically retrieve only the most pertinent blocks or tokens during decoding Ribar et al. (2023); Tang et al. (2024b); Chen et al. (2025); Yuan et al. (2025). This paradigm can effectively enhance throughput on extended sequences while closely approximating the accuracy of full-attention mechanisms. For instance, SparQ Ribar et al. (2023) estimates token significance via cache channel pruning, thereby facilitating the selection of salient tokens. Quest Tang et al. (2024b) query the min and max row of key cache to assess the criticality of token blocks. HShare (Wu et al., 2025) further facilitates critical KV cache token sharing when selecting cache. NSA (Yuan et al., 2025) extends the principle of KV selection to the pretraining phase, demonstrating the effectiveness of native attention sparsity. Nonetheless, cache selection techniques are primarily oriented towards accelerating the decoding process without alleviating the KV memory footprint, *i.e.*, they do not evict any KV cache but only retrieve cache for efficient computation, and thus fall beyond the scope of this paper.

## 4.3 Efficient Architecture

This line of work modifies the vanilla dense-attention architecture to natively improve KV-cache efficiency. For example, GQA (Ainslie et al., 2023) shares a common KV cache across queries from different heads, substantially reducing storage and access overhead; MLA (Liu et al., 2024a) introduces low-rank joint key–value compression that maps the KV cache to compact latent vectors. These architectural techniques are orthogonal to eviction/selection strategies and can often be composed to yield end-to-end gains in memory usage, latency, and throughput.

## 5 Conclusion

KV eviction is pivotal for efficient long-context LLMs. However, prevailing offline and online policies struggle to reconcile task generalization, deployment efficiency, and accuracy retention. We demonstrate that straightforward supervised fine-tuning (SFT) substantially closes the offline-to-online accuracy gap while preserving offline efficiency and robustness to multi-turn dialogue. We further introduce HS-SFT, which learns discrete, layer-wise local-window assignments for streaming heads via straight-through estimator and incorporates a budget-aware balancing loss to optimize the sparsity allocations at high compression rates. Across validation across an array of models and tasks, HS-SFT consistently surpasses state-of-the-art eviction baselines while retaining offline advantages, opening a training-centric alternative to hand-crafted pruning metrics for the community. Future work includes scaling SFT and exploring finer-grained, head-wise hybrid paradigms with hardware co-design.

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

# A APPENDIX

## A.1 USE OF LARGE LANGUAGE MODELS (LLMs)

During the writing of this paper, we utilized LLM solely for language editing to improve clarity and readability. We critically reviewed and revised all AI-generated suggestions to ensure the final text accurately reflects our original intent. All intellectual contributions, including the research design, methodology, analysis, and conclusions, are our exclusive work, and we take full responsibility for the academic integrity of this publication.

## A.2 DISCUSSION AND LIMITATION

In this section, we discuss the limitations and potential future work of HS-SFT. First, while our layer-specific router offers superior training and inference efficiency, head-specific KV sparsity (Ge et al., 2024) has more intuitive potential for higher task performance. A promising direction is to pair infrastructure innovations with head-specific hybrid configuration for SFT and inference. Second, our comparison against OSS-like coarse-grained hybrids is currently limited to the SFT setting, demonstrating HS-SFT as a scalable approach for rapidly transferring dense models to the sparse KV paradigm. An important next step is to evaluate the method during pretraining to assess upstream performance. Finally, our choice of dense heads follows conclusions from Duo-att (Xiao et al., 2025), and the sparsity ratio is largely fixed. Future work could explore a once-for-all hybrid configuration learned during training, together with flexible sparsity control at inference time.

## A.3 HS-SFT ALGORITHM WORKFLOW

We provide pseudocode for the training workflow of HS-SFT in Algorithm 1. For the inference workflow, we initialize per-layer KV budgets to their learned values. Then, We monotonically rescale only the sparse-head budgets so that their sum with the dense-head contribution exactly matches the global KV budget. Accordingly, dense heads attend to the full cache, whereas sparse heads attend to their rescaled per-layer budgets during inference.

---

**Algorithm 1** HS-SFT Training

---

**Inputs:** Pretrained LLM $\theta_0$, SFT dataset $\mathcal{D}$, dense-head fraction $\alpha$, budget set $\mathcal{B} = [b_1, \ldots, b_m]$, temperature $\tau$, prior-shape $\gamma$, balance weight $\lambda$, stability $\varepsilon$, optimizer $\mathrm{Opt}$, total steps $T$
**Outputs:** Fine-tuned weights $\theta^\star$, budget logits $\{\mathbf{z}\}$

1: Initialize dense heads per layer using Duo-Attn; initialize logits $\mathbf{z} \in \mathbb{R}^m$ for each sparse head
2: **for** $t = 1$ to $T$ **do**
3:      Sample minibatch $(x, y) \sim \mathcal{D}$
4:      **for** each layer and sparse head **do**
5:          $i^\star \leftarrow \arg\max_i z_i$; $k \leftarrow \mathcal{B}[i^\star]$                                      // Equation. 1
6:          Sparse attention computation with the 128 sink tokens and the latest $k$ tokens
7:      **end for**
8:      Compute $\mathcal{L}_{\mathrm{LM}}(\theta; x, y)$
9:      $p_i \leftarrow \mathrm{softmax}(\mathbf{z}/\tau)_i$                                              // Equation. 3
10:      $q_i \leftarrow \frac{(m+1-i)^\gamma}{\sum_{j=1}^{m}(m+1-j)^\gamma}$                        // Equation. 4
11:      $\mathcal{L}_{\mathrm{balance}} \leftarrow \lambda \sum_i p_i(\log(p_i + \varepsilon) - \log(q_i + \varepsilon))$       // Equation. 5
12:      $\mathcal{L} \leftarrow \mathcal{L}_{\mathrm{LM}} + \mathcal{L}_{\mathrm{balance}}$                                     // Equation. 6
13:      Backpropagate $\nabla_\theta \mathcal{L}$ and STE gradients for $\mathbf{z}$ as in Equation 2
14:      Update $(\theta, \{\mathbf{z}\}) \leftarrow \mathrm{Opt}$ step
15: **end for**
16: **return** $(\theta^\star, \{\mathbf{z}\})$

---

## A.4 DETAILED EXPERIMENT SETTINGS

We detail the hyperparameters used for HS-SFT in Table 6. The SFT settings are generally kept the same as Gao et al. (2024); Bhaskar et al. (2025).

Table 6: Detailed Hyperparameters used for HS-SFT.

| Hyperparameter | Value |
|---|---|
| *SFT base settings* | |
| Batch size (tokens) | 4,194,304 |
| Learning rate | $2 \cdot 10^{-5}$ |
| Training steps | 2500 |
| LR schedule | Linear warmup for first 5% of steps then linear decay to 10% peak LR |
| Adam $(\beta_1, \beta_2)$ | $(0.9, 0.95)$ |
| *HS-SFT hyperparameters* | |
| Logit LR | $3 \cdot 10^{-4}$ |
| Lambda | $1 \cdot 10^{-2}$ |
| Gamma | 2.0 |
| $\varepsilon$ | $1 \cdot 10^{-8}$ |
| Candidate budget set | $\{1, 2, 3, 4, 5, 6, 7, 8\} \times 128$ |
| Sink size | 128 |

## A.5 BUDGET ALLOCATION ANALYSIS

Figures 7 and 8 show the learned budget allocation across different SFT corpora and candidate budget sets, respectively. First, the budget allocation remains consistent across various SFT data settings, aligning with the results reported in Table 4. In contrast, varying the candidate budget set can lead to notable performance degradation, particularly when the budget search space is excessively large. This stems from the fact that modern SFT data typically has an average sequence length of approximately 1K tokens (Lambert et al., 2024; Ding et al., 2023). Consequently, if the learnable budget is set too high, *e.g.*, 32 blocks corresponding to a maximum budget of $32 \times 128 = 4096$, which means the model undergoes dense fine-tuning. This diverges from our primary objective of employing fine-tuning to mitigate performance losses caused by KV eviction. However, we observe that despite changes in budget candidates, the overall relative trend of the learned block sizes remains stable, underscoring the robustness of our proposed budget learning strategy.

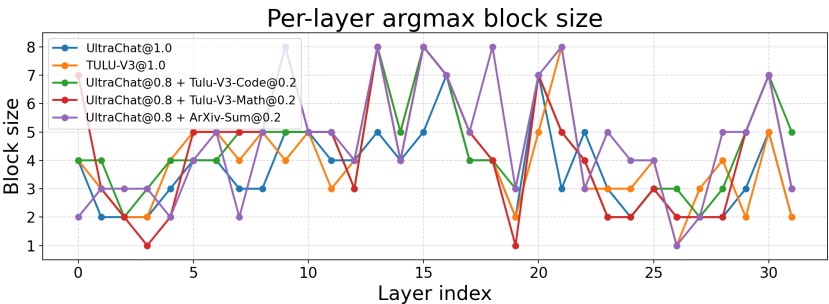

Figure 7: Learned budget allocation (blocks) across transformer layers of Llama-3-8B-1048K given different SFT corpus. Layer indices increase from left to right.

## A.6 EFFICIENCY ANALYSIS

We analyze the computational efficiency of HS-SFT in comparison to traditional KV eviction and various hybrid paradigms. As detailed in Section 2, HS-SFT partitions each layer into dense-head

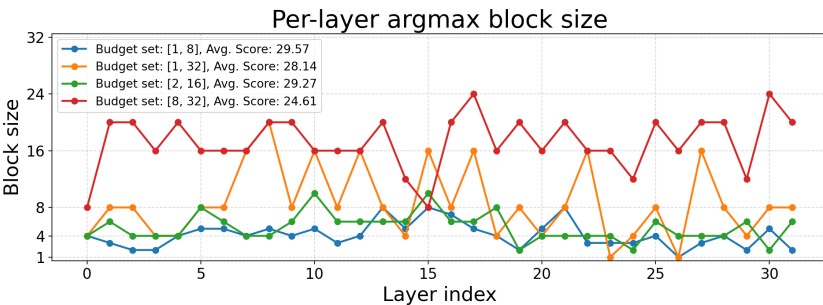

Figure 8: Learned budget allocation (blocks) across transformer layers of Llama-3-8B-1048K given different budget sets. Layer indices increase from left to right.

Table 7: Runtime efficiency under a fixed KV budget of Llama-3-8B-1048K. Per-layer hybrid refers to Duo-attention and our HS-SFT that divides heads of each layer into dense and streaming heads with uniform budgets.

| Method | Average Latency (ms) ↓ | Peak Memory (MB) ↓ | Speedup (↑) | Memory Reduction (%) ↑ |
|---|---|---|---|---|
| Dense | 113.26 | 47827.22 | 1.00× | 0.00 |
| Uniform | 34.82 | 18741.30 | 3.25× | 60.82 |
| Layer-wise hyrbid | 33.79 | 19980.01 | 3.35× | 58.24 |
| Per-layer hybrid | 37.59 | 18790.64 | 3.01× | 60.73 |
| Head-wise hybrid | 73.96 | 18923.39 | 1.53× | 60.43 |

and sparse-head flows, enforcing a shared budget across all sparse heads within a layer. Crucially, this design circumvents dependencies on specialized kernel design, thereby preserving high inference efficiency and deployment practicability. To substantiate this, we benchmark practical speedups using the native Transformers library under a fixed KV budget. We compare four distinct categories: (a) uniform KV eviction (*e.g.*, H2O and SnapKV); (b) layer-wise hybrid eviction (*e.g.*, GPT-OSS); (c) intra-layer sparse–dense hybrid strategies (*e.g.*, Duo-attn and HS-SFT); and (d) per-head sparsity with distinct rates (*e.g.*, HeadKV Fu et al. (2024b)). The results are summarized in Table 7. It is worth noting that variable-length FlashAttention techniques Dao et al. (2022); Feng et al. (2024) can support head-wise computation to achieve better acceleration effects. Specifically, one could leverage a custom CUDA kernel utilizing similar variable-length principles to attain speeds approaching conventional FlashAttention. This implies that with such kernel support, the acceleration effects of all methods in Table 7 would be consistent. However, operations requiring such specialized operator support are not applicable to the training scenarios investigated in this work and would necessitate more complex infrastructure in large-scale and practical settings. In contrast, HS-SFT achieves favorable deployment acceleration without reliance on any custom kernels.

Operationally, given a per-layer sparse budget, HS-SFT necessitates only two forward passes per layer—one for dense heads and one for sparse heads—thereby eliminating the need for granular per-head KV management. In contrast, purely head-wise approaches must manage KV selection and inference at the individual head level, which introduces significant overhead and often yields only marginal practical speedups. While specialized kernels can accelerate head-wise inference Fu et al. (2024b), such designs are difficult to generalize to the training phase. Ultimately, HS-SFT achieves a favorable balance between training efficiency and downstream performance, while remaining simple to implement and deploy.

## A.7 COMPARISON WITH ONLINE EVICTION METHODS

Table 8 presents a detailed comparison among HS-SFT, H2O Zhang et al. (2023b), MorphKV Ghadia et al. (2025), SnapKV Li et al. (2024), AdaKV Feng et al. (2024), and HeadKV Fu et al. (2024b) under a unified 50% KV cache budget on LongBench using the Llama-3-8B-Instruct-Gradient-1048k model. HS-SFT still holds robust performance strengths compared with online eviction methods, along with its unique benefits in pre-filling acceleration.

Table 8: Full LongBench results compared with online eviction methods at 50% KV budget for Llama-3-8B-Instruct-Gradient-1048k.

| Dataset | Full | H2O | MorphKV | SnapKV | AdaKV | HeadKV | HS-SFT |
|---|---|---|---|---|---|---|---|
| Average | 40.08 | 26.84 | 38.19 | 38.47 | 38.67 | 39.12 | 42.17 |
| 2WikiMQA | 28.78 | 28.87 | 24.01 | 29.00 | 28.97 | 29.01 | 30.60 |
| DuReader (zh) | 30.41 | 15.56 | 26.12 | 24.04 | 22.65 | 22.87 | 31.74 |
| GovReport | 34.23 | 20.66 | 27.19 | 26.84 | 24.22 | 24.01 | 32.60 |
| HotpotQA | 40.37 | 39.60 | 39.66 | 40.86 | 40.23 | 39.98 | 38.62 |
| LCC | 38.19 | 45.78 | 43.87 | 38.83 | 39.67 | 41.02 | 43.25 |
| LSHT (zh) | 38.00 | 16.50 | 35.00 | 38.00 | 36.50 | 37.14 | 34.50 |
| MultiNews | 27.73 | 19.21 | 28.40 | 22.84 | 21.81 | 22.04 | 27.69 |
| MultiFieldQA-en | 52.62 | 21.01 | 50.57 | 51.96 | 52.99 | 53.01 | 53.34 |
| MultiFieldQA-zh | 50.58 | 19.81 | 51.12 | 50.74 | 50.59 | 50.88 | 51.28 |
| Musique | 24.22 | 20.63 | 18.82 | 24.86 | 24.68 | 24.91 | 15.95 |
| NarrativeQA | 26.56 | 19.14 | 22.69 | 26.63 | 27.36 | 27.92 | 27.75 |
| Passage Count | 1.00 | 0.53 | 1.00 | 1.00 | 1.00 | 1.00 | 1.50 |
| PassageRetrieval-en | 81.00 | 19.50 | 81.00 | 80.50 | 80.50 | 83.00 | 92.50 |
| PassageRetrieval-zh | 62.15 | 11.75 | 60.10 | 58.53 | 61.92 | 62.45 | 88.56 |
| Qasper | 29.21 | 16.84 | 23.16 | 26.00 | 27.02 | 28.34 | 37.23 |
| QMSum | 24.52 | 18.89 | 23.82 | 24.90 | 24.65 | 24.55 | 24.86 |
| RepoBench-P | 38.94 | 45.16 | 42.33 | 38.20 | 38.50 | 39.12 | 40.80 |
| SAMSum | 42.51 | 39.73 | 39.93 | 40.90 | 41.38 | 41.88 | 42.31 |
| TREC | 71.50 | 48.50 | 64.00 | 66.00 | 71.00 | 71.00 | 71.00 |
| TriviaQA | 87.70 | 85.16 | 88.21 | 87.30 | 86.80 | 87.43 | 88.39 |
| VCSUM (zh) | 11.37 | 10.71 | 11.02 | 9.91 | 9.62 | 9.99 | 11.17 |

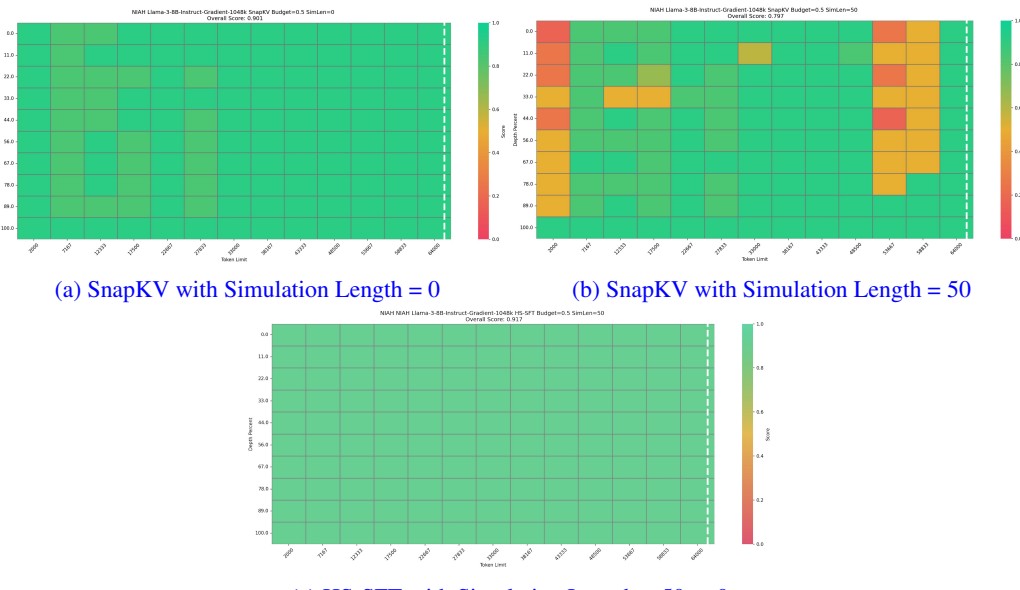

(a) SnapKV with Simulation Length = 0

(b) SnapKV with Simulation Length = 50

(c) HS-SFT with Simulation Length = 50 or 0

Figure 9: NIAH results for Llama-3-8B-1048K with a 50% KV cache budget.

Moreover, we posit that offline KV eviction is significantly more robust in real-world scenarios, such as multi-round dialogue, where the query is not necessarily located at the end of the context. To validate this, we follow prior works (Xiao et al., 2025; Tang et al., 2024b) in evaluating KV cache eviction methods on a variant of the NIAH benchmark. In this setting, the final 50 tokens of the prompt act as simulated generated output, mimicking a second-round dialogue scenario. As illustrated in Figure 9, while SnapKV correctly retrieves the answer when the query is adjacent to the end, its

performance collapses when the query position shifts. In contrast, HS-SFT remains stable across test scenarios, as the offline eviction strategy stays invariant to the simulated generation process. These findings underscore the applicability of offline eviction methods to real-world retrieval and multi-turn interactions.

## A.8 RESULTS ON QWEN-2.5-32B MODEL

We further validate HS-SFT on models beyond the LLaMA family by evaluating Qwen-2.5-32B Team et al. (2024), a strong open-source LLM with 128k context window. Table 9 shows that HS-SFT maintains a consistent advantage on Qwen-2.5-32B under 50% KV budget.

Table 9: Full LongBench results at 50% KV budget for Qwen-2.5-32B.

| Dataset | Baseline | SLLM | Duo | HS-SFT |
| --- | --- | --- | --- | --- |
| Average | 44.53 | 36.42 | 43.42 | 44.31 |
| 2WikiMQA | 39.96 | 37.96 | 39.89 | 38.01 |
| DuReader (zh) | 30.29 | 24.34 | 28.84 | 29.96 |
| GovReport | 35.57 | 33.75 | 34.65 | 35.14 |
| HotpotQA | 46.89 | 41.56 | 48.51 | 46.03 |
| LCC | 48.47 | 45.20 | 44.62 | 47.18 |
| LSHT (zh) | 48.50 | 38.25 | 43.50 | 48.00 |
| MultiNews | 24.37 | 23.77 | 24.55 | 24.22 |
| MultiFieldQA-en | 42.94 | 29.30 | 40.34 | 42.51 |
| MultiFieldQA-zh | 61.99 | 37.69 | 62.03 | 61.43 |
| Musique | 28.59 | 25.31 | 28.47 | 26.45 |
| NarrativeQA | 21.02 | 16.77 | 20.21 | 21.53 |
| Passage Count | 14.12 | 9.22 | 13.02 | 14.51 |
| PassageRetrieval-en | 95.25 | 56.17 | 93.51 | 95.59 |
| PassageRetrieval-zh | 92.77 | 52.71 | 92.08 | 93.12 |
| Qasper | 20.85 | 18.96 | 19.01 | 21.23 |
| QMSum | 23.41 | 20.72 | 23.88 | 24.12 |
| RepoBench-P | 32.57 | 34.49 | 33.41 | 35.88 |
| SAMSum | 47.12 | 45.61 | 46.42 | 46.99 |
| TREC | 72.50 | 69.50 | 72.50 | 71.50 |
| TriviaQA | 88.54 | 85.99 | 86.72 | 88.72 |
| VCSUM (zh) | 18.36 | 17.55 | 15.75 | 18.44 |

## A.9 DETAILED QUANTITATIVE RESULTS

We provide the detailed quantitative results of each task on LongBench and Ruler in the following tables.

Table 10: Full LongBench results at 50% KV budget for Llama-3-8.1B-Instruct.

| Dataset | Full | H2O | SLLM | Duo | HS-SFT |
|---|---|---|---|---|---|
| Average | 39.01 | 35.61 | 31.32 | 38.91 | 41.85 |
| 2WikiMQA | 16.37 | 13.91 | 13.25 | 16.20 | 17.74 |
| DuReader (zh) | 29.30 | 21.53 | 12.95 | 31.31 | 30.76 |
| GovReport | 34.53 | 30.56 | 30.47 | 32.87 | 33.89 |
| HotpotQA | 17.23 | 17.31 | 15.78 | 19.53 | 20.49 |
| LCC | 52.39 | 53.08 | 52.90 | 53.31 | 48.97 |
| LSHT (zh) | 46.00 | 39.00 | 36.00 | 45.00 | 41.50 |
| MultiNews | 26.91 | 25.52 | 24.97 | 26.29 | 27.63 |
| MultiFieldQA-en | 28.44 | 21.89 | 16.05 | 27.77 | 33.66 |
| MultiFieldQA-zh | 20.19 | 14.87 | 15.92 | 21.98 | 60.83 |
| Musique | 11.82 | 10.15 | 10.19 | 12.97 | 14.25 |
| NarrativeQA | 31.99 | 31.09 | 24.15 | 29.12 | 28.77 |
| Passage Count | 6.26 | 5.40 | 4.75 | 6.31 | 2.00 |
| PassageRetrieval-en | 97.95 | 89.86 | 52.11 | 98.59 | 96.50 |
| PassageRetrieval-zh | 77.54 | 69.73 | 35.14 | 75.37 | 94.00 |
| Qasper | 25.14 | 16.96 | 23.56 | 21.12 | 28.47 |
| QMSum | 23.63 | 22.54 | 21.48 | 23.89 | 26.37 |
| RepoBench-P | 49.46 | 49.51 | 49.95 | 53.74 | 52.45 |
| SAMSum | 43.69 | 42.56 | 43.32 | 43.40 | 42.06 |
| TREC | 72.50 | 66.50 | 69.50 | 73.00 | 74.50 |
| TriviaQA | 91.65 | 90.07 | 90.06 | 89.60 | 87.99 |
| VCSUM (zh) | 16.26 | 15.80 | 15.17 | 15.83 | 16.01 |

Table 11: Full LongBench results at 50% KV budget for Llama-3-8B-Instruct-262K.

| Dataset | Baseline | SLLM | Duo | HS-SFT |
|---|---|---|---|---|
| Average | 41.11 | 33.48 | 41.83 | 42.33 |
| 2WikiMQA | 24.43 | 24.15 | 29.27 | 26.71 |
| DuReader (zh) | 31.94 | 28.55 | 32.39 | 31.98 |
| GovReport | 34.69 | 30.48 | 32.89 | 33.90 |
| HotpotQA | 38.22 | 30.53 | 38.84 | 31.80 |
| LCC | 43.81 | 45.05 | 49.85 | 48.87 |
| LSHT (zh) | 43.00 | 31.50 | 41.00 | 35.25 |
| MultiNews | 27.15 | 25.42 | 27.88 | 27.23 |
| MultiFieldQA-en | 46.19 | 33.08 | 46.23 | 52.99 |
| MultiFieldQA-zh | 52.03 | 31.86 | 54.06 | 50.74 |
| Musique | 19.94 | 17.24 | 19.19 | 15.25 |
| NarrativeQA | 23.94 | 21.42 | 23.55 | 27.93 |
| Passage Count | 0.00 | 0.5 | 0.0 | 1.50 |
| PassageRetrieval-en | 87.00 | 47.5 | 85.0 | 85.00 |
| PassageRetrieval-zh | 76.76 | 39.33 | 79.95 | 93.00 |
| Qasper | 33.24 | 24.35 | 30.64 | 36.64 |
| QMSum | 25.45 | 23.03 | 26.20 | 25.54 |
| RepoBench-P | 40.54 | 41.84 | 48.02 | 46.98 |
| SAMSum | 41.75 | 40.20 | 40.19 | 43.15 |
| TREC | 69.50 | 68.50 | 72.50 | 73.50 |
| TriviaQA | 88.19 | 85.16 | 85.91 | 85.77 |
| VCSUM (zh) | 15.49 | 13.37 | 14.80 | 15.10 |

Table 12: Full LongBench results at 30% KV budget for Llama-3-8B-Instruct-262K.

| Dataset | Baseline | SLLM | Duo | HS-SFT |
|---|---|---|---|---|
| Average | 41.11 | 29.94 | 38.81 | 41.30 |
| 2WikiMQA | 24.43 | 24.40 | 26.14 | 25.04 |
| DuReader (zh) | 31.94 | 28.09 | 32.08 | 33.11 |
| GovReport | 34.69 | 28.12 | 27.08 | 31.10 |
| HotpotQA | 38.22 | 28.05 | 34.42 | 34.88 |
| LCC | 43.81 | 43.43 | 46.92 | 49.78 |
| LSHT (zh) | 43.00 | 27.00 | 32.00 | 32.50 |
| MultiNews | 27.15 | 24.07 | 26.75 | 27.85 |
| MultiFieldQA-en | 46.19 | 32.16 | 42.18 | 52.90 |
| MultiFieldQA-zh | 52.03 | 26.81 | 53.96 | 49.44 |
| Musique | 19.94 | 15.51 | 16.80 | 15.58 |
| NarrativeQA | 23.94 | 20.90 | 21.69 | 24.41 |
| Passage Count | 0.00 | 1.50 | 1.00 | 3.50 |
| PassageRetrieval-en | 87.00 | 37.00 | 85.00 | 85.00 |
| PassageRetrieval-zh | 76.76 | 22.77 | 66.54 | 85.56 |
| Qasper | 33.24 | 17.04 | 31.39 | 36.41 |
| QMSum | 25.45 | 21.28 | 24.04 | 24.35 |
| RepoBench-P | 40.54 | 42.10 | 45.70 | 49.00 |
| SAMSum | 41.75 | 37.61 | 38.57 | 41.05 |
| TREC | 69.50 | 57.50 | 67.50 | 72.00 |
| TriviaQA | 88.19 | 80.27 | 79.87 | 78.12 |
| VCSUM (zh) | 15.49 | 13.18 | 15.43 | 15.62 |

Table 13: Full LongBench results at 10% KV budget for Llama-3-8B-Instruct-262K.

| Dataset | Baseline | SLLM | Duo | HS-SFT |
|---|---|---|---|---|
| Average | 41.11 | 24.20 | 24.53 | 29.38 |
| 2WikiMQA | 24.43 | 22.34 | 22.13 | 23.94 |
| DuReader (zh) | 31.94 | 28.10 | 29.40 | 33.89 |
| GovReport | 34.69 | 22.79 | 20.92 | 26.44 |
| HotpotQA | 38.22 | 22.92 | 25.78 | 23.32 |
| LCC | 43.81 | 42.55 | 43.82 | 47.98 |
| LSHT (zh) | 43.00 | 17.75 | 11.00 | 22.25 |
| MultiNews | 27.15 | 22.73 | 23.58 | 26.30 |
| MultiFieldQA-en | 46.19 | 28.80 | 30.38 | 39.91 |
| MultiFieldQA-zh | 52.03 | 22.08 | 25.34 | 34.31 |
| Musique | 19.94 | 12.46 | 11.83 | 10.14 |
| NarrativeQA | 23.94 | 18.23 | 17.64 | 19.54 |
| Passage Count | 0.00 | 1.00 | 2.00 | 3.00 |
| PassageRetrieval-en | 87.00 | 12.50 | 9.50 | 14.50 |
| PassageRetrieval-zh | 76.76 | 6.72 | 8.64 | 15.88 |
| Qasper | 33.24 | 13.34 | 14.18 | 21.95 |
| QMSum | 25.45 | 19.29 | 20.86 | 22.40 |
| RepoBench-P | 40.54 | 40.16 | 39.02 | 42.11 |
| SAMSum | 41.75 | 36.58 | 35.95 | 39.76 |
| TREC | 69.50 | 35.00 | 42.00 | 57.50 |
| TriviaQA | 88.19 | 72.04 | 69.16 | 77.32 |
| VCSUM (zh) | 15.49 | 10.81 | 12.01 | 14.58 |

Table 14: Full LongBench results at 50% KV budget for Llama-3-8B-1048K.

| Dataset | Baseline | SLLM | Duo | HS-SFT |
|---|---|---|---|---|
| Average | 40.09 | 32.26 | 39.45 | 42.17 |
| 2WikiMQA | 28.78 | 22.64 | 29.61 | 30.60 |
| DuReader (zh) | 30.41 | 25.85 | 28.99 | 31.74 |
| GovReport | 34.23 | 30.08 | 32.28 | 32.60 |
| HotpotQA | 40.37 | 34.84 | 43.37 | 38.62 |
| LCC | 38.19 | 40.08 | 37.71 | 43.25 |
| LSHT (zh) | 38.00 | 25.50 | 30.50 | 34.50 |
| MultiNews | 27.73 | 25.49 | 27.93 | 27.69 |
| MultiFieldQA-en | 52.62 | 32.46 | 50.12 | 53.34 |
| MultiFieldQA-zh | 50.98 | 32.24 | 52.54 | 51.28 |
| Musique | 24.22 | 19.65 | 24.39 | 15.95 |
| NarrativeQA | 26.56 | 20.71 | 25.35 | 27.75 |
| Passage Count | 1.00 | 1.00 | 1.00 | 1.50 |
| PassageRetrieval-en | 81.00 | 46.50 | 83.50 | 92.50 |
| PassageRetrieval-zh | 62.15 | 34.21 | 60.41 | 88.56 |
| Qasper | 29.21 | 18.81 | 27.51 | 37.23 |
| QMSum | 24.52 | 22.56 | 24.72 | 24.86 |
| RepoBench-P | 38.94 | 39.83 | 39.73 | 40.80 |
| SAMSum | 42.51 | 40.42 | 41.89 | 42.31 |
| TREC | 71.50 | 68.00 | 73.00 | 71.00 |
| TriviaQA | 87.70 | 84.72 | 86.69 | 88.39 |
| VCSUM (zh) | 11.37 | 11.93 | 7.15 | 11.17 |

Table 15: Full LongBench results at 30% KV budget for Llama-3-8B-1048K.

| Dataset | Baseline | SLLM | Duo | HS-SFT |
|---|---|---|---|---|
| Average | 40.09 | 29.54 | 30.28 | 39.22 |
| 2WikiMQA | 28.78 | 24.10 | 23.15 | 26.48 |
| DuReader (zh) | 30.41 | 25.85 | 26.85 | 32.79 |
| GovReport | 34.23 | 28.07 | 22.19 | 29.03 |
| HotpotQA | 40.37 | 29.15 | 31.15 | 35.11 |
| LCC | 38.19 | 42.67 | 35.18 | 47.24 |
| LSHT (zh) | 38.00 | 24.00 | 22.50 | 28.00 |
| MultiNews | 27.73 | 23.63 | 24.27 | 26.72 |
| MultiFieldQA-en | 52.62 | 31.92 | 41.06 | 52.13 |
| MultiFieldQA-zh | 50.98 | 25.62 | 39.70 | 45.77 |
| Musique | 24.22 | 18.34 | 10.52 | 13.68 |
| NarrativeQA | 26.56 | 20.54 | 18.64 | 25.22 |
| Passage Count | 1.00 | 1.50 | 1.50 | 1.50 |
| PassageRetrieval-en | 81.00 | 35.50 | 49.50 | 78.50 |
| PassageRetrieval-zh | 62.15 | 20.59 | 27.75 | 67.50 |
| Qasper | 29.21 | 14.61 | 18.48 | 36.97 |
| QMSum | 24.52 | 21.00 | 22.29 | 23.36 |
| RepoBench-P | 38.94 | 41.67 | 37.54 | 42.89 |
| SAMSum | 42.51 | 38.90 | 40.98 | 42.22 |
| TREC | 71.50 | 59.00 | 57.50 | 68.50 |
| TriviaQA | 87.70 | 82.55 | 78.06 | 87.97 |
| VCSUM (zh) | 11.37 | 11.22 | 7.02 | 11.96 |

Table 16: Full LongBench results at 10% KV budget for Llama-3-8B-1048K.

| Dataset | Baseline | SLLM | Duo | HS-SFT |
|---|---|---|---|---|
| Average | 40.09 | 24.58 | 23.72 | 29.57 |
| 2WikiMQA | 28.78 | 20.08 | 19.45 | 26.64 |
| DuReader (zh) | 30.41 | 27.47 | 28.42 | 28.40 |
| GovReport | 34.23 | 22.49 | 20.85 | 27.06 |
| HotpotQA | 40.37 | 24.47 | 26.21 | 30.12 |
| LCC | 38.19 | 42.10 | 43.12 | 47.50 |
| LSHT (zh) | 38.00 | 16.25 | 14.50 | 18.25 |
| MultiNews | 27.73 | 22.36 | 23.10 | 25.23 |
| MultiFieldQA-en | 52.62 | 29.66 | 27.97 | 34.66 |
| MultiFieldQA-zh | 50.98 | 24.27 | 24.22 | 29.79 |
| Musique | 24.22 | 12.93 | 9.63 | 11.17 |
| NarrativeQA | 26.56 | 19.92 | 16.20 | 20.48 |
| Passage Count | 1.00 | 1.00 | 0.50 | 1.50 |
| PassageRetrieval-en | 81.00 | 14.00 | 7.55 | 18.50 |
| PassageRetrieval-zh | 62.15 | 6.30 | 5.50 | 18.50 |
| Qasper | 29.21 | 11.17 | 11.12 | 25.22 |
| QMSum | 24.52 | 19.39 | 19.19 | 22.28 |
| RepoBench-P | 38.94 | 41.25 | 42.30 | 43.24 |
| SAMSum | 42.51 | 37.04 | 36.26 | 41.10 |
| TREC | 71.50 | 37.50 | 38.00 | 58.00 |
| TriviaQA | 87.70 | 76.66 | 73.47 | 81.67 |
| VCSUM (zh) | 11.37 | 9.79 | 10.50 | 11.73 |

Table 17: Full LongBench results at 50% KV budget for Llama-2-32K.

| Dataset | Baseline | SLLM | Duo | HS-SFT |
|---|---|---|---|---|
| Average | 37.53 | 32.75 | 36.80 | 36.81 |
| 2WikiMQA | 35.59 | 31.51 | 37.30 | 26.34 |
| DuReader (zh) | 25.10 | 18.65 | 24.00 | 27.84 |
| GovReport | 31.19 | 26.42 | 30.37 | 31.79 |
| HotpotQA | 47.98 | 44.98 | 48.84 | 41.95 |
| LCC | 51.21 | 48.36 | 49.28 | 49.34 |
| LSHT (zh) | 34.50 | 27.50 | 32.00 | 32.00 |
| MultiNews | 27.14 | 24.94 | 26.07 | 27.66 |
| MultiFieldQA-en | 33.95 | 21.35 | 34.29 | 31.86 |
| MultiFieldQA-zh | 45.79 | 30.17 | 46.76 | 50.10 |
| Musique | 22.97 | 22.01 | 20.81 | 18.54 |
| NarrativeQA | 24.11 | 22.83 | 23.58 | 22.17 |
| Passage Count | 0.00 | 0.85 | 0.00 | 0.12 |
| PassageRetrieval-en | 50.92 | 36.33 | 45.92 | 47.25 |
| PassageRetrieval-zh | 37.68 | 28.84 | 42.81 | 48.00 |
| Qasper | 33.23 | 28.19 | 30.32 | 32.52 |
| QMSum | 20.81 | 19.68 | 20.59 | 23.22 |
| RepoBench-P | 51.58 | 49.96 | 48.91 | 48.34 |
| SAMSum | 42.10 | 40.15 | 42.07 | 41.92 |
| TREC | 71.50 | 66.00 | 70.50 | 71.50 |
| TriviaQA | 86.21 | 86.71 | 85.91 | 87.27 |
| VCSUM (zh) | 14.51 | 12.25 | 12.42 | 13.34 |

Table 18: Full LongBench results at 30% KV budget for Llama-2-32K.

| Dataset | Baseline | SLLM | Duo | HS-SFT |
|---|---|---|---|---|
| Average | 37.53 | 29.93 | 36.16 | 36.10 |
| 2WikiMQA | 35.59 | 29.49 | 32.80 | 28.68 |
| DuReader (zh) | 25.10 | 18.52 | 23.78 | 27.76 |
| GovReport | 31.19 | 24.53 | 29.99 | 31.55 |
| HotpotQA | 47.98 | 40.29 | 48.36 | 42.72 |
| LCC | 51.21 | 48.20 | 49.01 | 50.02 |
| LSHT (zh) | 34.50 | 24.00 | 31.50 | 26.50 |
| MultiNews | 27.14 | 23.96 | 26.46 | 27.00 |
| MultiFieldQA-en | 33.95 | 19.28 | 30.38 | 29.87 |
| MultiFieldQA-zh | 45.79 | 25.28 | 40.79 | 47.95 |
| Musique | 22.97 | 19.57 | 19.94 | 19.77 |
| NarrativeQA | 24.11 | 20.50 | 21.98 | 19.65 |
| Passage Count | 0.00 | 0.50 | 0.42 | 0.12 |
| PassageRetrieval-en | 50.92 | 26.42 | 52.63 | 45.42 |
| PassageRetrieval-zh | 37.68 | 20.18 | 48.27 | 44.25 |
| Qasper | 33.23 | 21.86 | 26.70 | 30.71 |
| QMSum | 20.81 | 19.55 | 21.19 | 22.26 |
| RepoBench-P | 51.58 | 48.99 | 48.95 | 50.35 |
| SAMSum | 42.10 | 38.50 | 37.01 | 41.29 |
| TREC | 71.50 | 62.00 | 71.00 | 71.00 |
| TriviaQA | 86.21 | 85.28 | 86.23 | 87.94 |
| VCSUM (zh) | 14.51 | 11.61 | 12.00 | 13.20 |

Table 19: Full LongBench results at 10% KV budget for Llama-2-32K.

| Dataset | Baseline | SLLM | Duo | HS-SFT |
|---|---|---|---|---|
| Average | 37.53 | 25.51 | 7.33 | 25.98 |
| 2WikiMQA | 35.59 | 25.02 | 10.75 | 18.62 |
| DuReader (zh) | 25.10 | 19.00 | 1.15 | 21.80 |
| GovReport | 31.19 | 22.14 | 5.73 | 23.27 |
| HotpotQA | 47.98 | 32.95 | 2.37 | 37.44 |
| LCC | 51.21 | 45.72 | 20.40 | 49.60 |
| LSHT (zh) | 34.50 | 14.75 | 0.00 | 16.50 |
| MultiNews | 27.14 | 20.97 | 13.29 | 23.79 |
| MultiFieldQA-en | 33.95 | 17.13 | 11.57 | 16.99 |
| MultiFieldQA-zh | 45.79 | 18.24 | 2.81 | 25.58 |
| Musique | 22.97 | 17.62 | 0.05 | 12.73 |
| NarrativeQA | 24.11 | 18.74 | 0.71 | 11.74 |
| Passage Count | 0.00 | 0.85 | 0.12 | 0.27 |
| PassageRetrieval-en | 50.92 | 8.75 | 3.93 | 5.25 |
| PassageRetrieval-zh | 37.68 | 10.79 | 0.33 | 9.92 |
| Qasper | 33.23 | 16.69 | 11.11 | 19.58 |
| QMSum | 20.81 | 20.02 | 8.28 | 20.35 |
| RepoBench-P | 51.58 | 45.76 | 15.36 | 46.57 |
| SAMSum | 42.10 | 35.13 | 7.30 | 39.88 |
| TREC | 71.50 | 51.00 | 22.00 | 54.50 |
| TriviaQA | 86.21 | 83.52 | 13.83 | 81.07 |
| VCSUM (zh) | 14.51 | 10.94 | 2.84 | 10.20 |

Table 20: Full Ruler results at 50% KV budget for Llama-3-8B-Instruct-262K.

| Dataset | Baseline | SLLM | Duo | HS-SFT |
|---|---|---|---|---|
| Average | 92.68 | 49.17 | 90.47 | 92.96 |
| niah_single_1 | 100.00 | 47.20 | 100.00 | 100.00 |
| niah_single_2 | 100.00 | 44.40 | 100.00 | 100.00 |
| niah_single_3 | 100.00 | 47.80 | 100.00 | 100.00 |
| niah_multikey_1 | 99.60 | 51.60 | 99.60 | 99.40 |
| niah_multikey_2 | 99.60 | 47.20 | 100.00 | 99.60 |
| niah_multikey_3 | 96.40 | 46.40 | 97.00 | 92.60 |
| niah_multiquer | 99.95 | 48.25 | 99.65 | 99.80 |
| niah_multivalue | 92.40 | 47.50 | 97.05 | 98.80 |
| cwe | 43.46 | 11.06 | 18.72 | 44.40 |
| fwe | 90.87 | 92.33 | 88.67 | 91.00 |
| vt | 97.24 | 57.08 | 94.44 | 96.96 |

Table 21: Full Ruler results at 30% KV budget for Llama-3-8B-Instruct-262K.

| Dataset | Baseline | SLLM | Duo | HS-SFT |
|---|---|---|---|---|
| Average | 92.68 | 32.35 | 84.96 | 85.91 |
| niah_single_1 | 100.00 | 28.80 | 100.00 | 100.00 |
| niah_single_2 | 100.00 | 24.60 | 100.00 | 99.80 |
| niah_single_3 | 100.00 | 28.00 | 99.80 | 98.40 |
| niah_multikey_1 | 99.60 | 31.20 | 98.40 | 98.00 |
| niah_multikey_2 | 99.60 | 29.80 | 98.80 | 99.60 |
| niah_multikey_3 | 96.40 | 26.00 | 62.00 | 67.60 |
| niah_multiquer | 99.95 | 28.55 | 96.90 | 93.20 |
| niah_multivalue | 92.40 | 27.50 | 91.85 | 94.50 |
| cwe | 43.46 | 2.64 | 3.20 | 8.18 |
| fwe | 90.87 | 94.80 | 89.07 | 95.27 |
| vt | 97.24 | 33.96 | 94.52 | 90.48 |

Table 22: Full Ruler results at 10% KV budget for Llama-3-8B-Instruct-262K.

| Dataset | Baseline | SLLM | Duo | HS-SFT |
|---|---|---|---|---|
| Average | 92.68 | 16.18 | 9.14 | 18.44 |
| niah_single_1 | 100.00 | 8.40 | 9.40 | 13.40 |
| niah_single_2 | 100.00 | 7.60 | 15.80 | 13.00 |
| niah_single_3 | 100.00 | 9.20 | 8.80 | 14.20 |
| niah_multikey_1 | 99.60 | 9.40 | 25.00 | 16.40 |
| niah_multikey_2 | 99.60 | 8.60 | 11.80 | 11.00 |
| niah_multikey_3 | 96.40 | 8.00 | 1.60 | 11.00 |
| niah_multiquer | 99.95 | 10.00 | 4.70 | 10.30 |
| niah_multivalue | 92.40 | 9.20 | 4.85 | 9.20 |
| cwe | 43.46 | 0.28 | 0.76 | 3.78 |
| fwe | 90.87 | 97.60 | 16.13 | 92.20 |
| vt | 97.24 | 9.72 | 1.68 | 8.40 |

Table 23: Full Ruler results at 50% KV budget for Llama-3-8B-1048K.

| Dataset | Baseline | SLLM | Duo | HS-SFT |
|---|---|---|---|---|
| Average | 92.68 | 49.17 | 90.47 | 93.87 |
| niah_single_1 | 100.00 | 47.20 | 100.00 | 100.00 |
| niah_single_2 | 100.00 | 44.40 | 100.00 | 100.00 |
| niah_single_3 | 100.00 | 47.80 | 100.00 | 100.00 |
| niah_multikey_1 | 99.60 | 51.60 | 99.60 | 99.20 |
| niah_multikey_2 | 99.60 | 47.20 | 100.00 | 99.20 |
| niah_multikey_3 | 96.40 | 46.40 | 97.00 | 98.00 |
| niah_multiquer | 99.95 | 48.25 | 99.65 | 99.65 |
| niah_multivalue | 92.40 | 47.50 | 97.05 | 98.75 |
| cwe | 43.46 | 11.06 | 18.72 | 50.14 |
| fwe | 90.87 | 92.33 | 88.67 | 91.67 |
| vt | 97.24 | 57.08 | 94.44 | 95.96 |

Table 24: Full Ruler results at 30% KV budget for Llama-3-8B-1048K.

| Dataset | Baseline | SLLM | Duo | HS-SFT |
|---|---|---|---|---|
| Average | 92.68 | 32.35 | 84.96 | 85.94 |
| niah_single_1 | 100.00 | 28.80 | 100.00 | 100.00 |
| niah_single_2 | 100.00 | 24.60 | 100.00 | 100.00 |
| niah_single_3 | 100.00 | 28.00 | 99.80 | 98.40 |
| niah_multikey_1 | 99.60 | 31.20 | 98.40 | 98.20 |
| niah_multikey_2 | 99.60 | 29.80 | 98.80 | 97.80 |
| niah_multikey_3 | 96.40 | 26.00 | 62.00 | 84.40 |
| niah_multiquer | 99.95 | 28.55 | 96.90 | 92.10 |
| niah_multivalue | 92.40 | 27.50 | 91.85 | 84.45 |
| cwe | 43.46 | 2.64 | 3.20 | 9.32 |
| fwe | 90.87 | 94.80 | 89.07 | 93.47 |
| vt | 97.24 | 33.96 | 94.52 | 87.16 |

Table 25: Full Ruler results at 10% KV budget for Llama-3-8B-1048K.

| Dataset | Baseline | SLLM | Duo | HS-SFT |
|---|---|---|---|---|
| Average | 92.68 | 16.18 | 16.41 | 54.69 |
| niah_single_1 | 100.00 | 8.40 | 9.40 | 99.40 |
| niah_single_2 | 100.00 | 7.60 | 15.80 | 96.80 |
| niah_single_3 | 100.00 | 9.20 | 8.80 | 91.60 |
| niah_multikey_1 | 99.60 | 9.40 | 25.00 | 68.80 |
| niah_multikey_2 | 99.60 | 8.60 | 11.80 | 47.40 |
| niah_multikey_3 | 96.40 | 8.00 | 1.60 | 3.00 |
| niah_multiquer | 99.95 | 10.00 | 4.70 | 42.45 |
| niah_multivalue | 92.40 | 9.20 | 4.85 | 34.65 |
| cwe | 43.46 | 0.28 | 0.76 | 1.38 |
| fwe | 90.87 | 97.60 | 96.13 | 97.07 |
| vt | 97.24 | 9.72 | 1.68 | 19.00 |

Table 26: Full Ruler results at 50% KV budget for Llama-2-7B-32K.

| Dataset | Baseline | SLLM | Duo | HS-SFT |
|---|---|---|---|---|
| Average | 26.42 | 18.02 | 26.44 | 26.34 |
| niah_single_1 | 22.60 | 14.80 | 22.20 | 22.40 |
| niah_single_2 | 22.40 | 10.20 | 22.40 | 22.40 |
| niah_single_3 | 21.60 | 11.80 | 22.00 | 22.20 |
| niah_multikey_1 | 22.20 | 13.40 | 22.20 | 21.20 |
| niah_multikey_2 | 16.00 | 9.20 | 16.00 | 15.80 |
| niah_multikey_3 | 12.60 | 7.60 | 12.40 | 9.80 |
| niah_multiquer | 21.15 | 12.75 | 21.15 | 21.40 |
| niah_multivalue | 21.20 | 12.75 | 21.20 | 21.20 |
| cwe | 18.50 | 23.66 | 18.14 | 16.64 |
| fwe | 84.13 | 70.33 | 84.93 | 88.07 |
| vt | 28.24 | 11.76 | 28.20 | 28.64 |

Table 27: Full Ruler results at 30% KV budget for Llama-2-7B-32K.

| Dataset | Baseline | SLLM | Duo | HS-SFT |
|---|---|---|---|---|
| Average | 26.42 | 14.70 | 23.89 | 25.11 |
| niah_single_1 | 22.60 | 8.40 | 19.80 | 22.20 |
| niah_single_2 | 22.40 | 7.60 | 21.60 | 22.20 |
| niah_single_3 | 21.60 | 8.80 | 18.60 | 22.20 |
| niah_multikey_1 | 22.20 | 9.40 | 19.40 | 22.00 |
| niah_multikey_2 | 16.00 | 6.60 | 12.00 | 15.80 |
| niah_multikey_3 | 12.60 | 5.40 | 7.60 | 8.00 |
| niah_multiquer | 21.15 | 10.00 | 18.80 | 21.25 |
| niah_multivalue | 21.20 | 9.35 | 19.15 | 20.95 |
| cwe | 18.50 | 14.82 | 14.76 | 10.00 |
| fwe | 84.13 | 74.93 | 87.07 | 88.87 |
| vt | 28.24 | 6.44 | 24.00 | 22.72 |

Table 28: Full Ruler results at 10% KV budget for Llama-2-7B-32K.

| Dataset | Baseline | SLLM | Duo | HS-SFT |
|---|---|---|---|---|
| Average | 26.42 | 10.69 | 3.14 | 19.15 |
| niah_single_1 | 22.60 | 3.00 | 0.80 | 18.40 |
| niah_single_2 | 22.40 | 2.80 | 7.80 | 18.80 |
| niah_single_3 | 21.60 | 3.40 | 0.20 | 22.00 |
| niah_multikey_1 | 22.20 | 5.20 | 7.80 | 19.40 |
| niah_multikey_2 | 16.00 | 3.20 | 0.00 | 6.80 |
| niah_multikey_3 | 12.60 | 2.60 | 0.00 | 1.80 |
| niah_multiquer | 21.15 | 4.60 | 4.95 | 17.50 |
| niah_multivalue | 21.20 | 4.80 | 6.55 | 16.95 |
| cwe | 18.50 | 6.82 | 0.00 | 5.30 |
| fwe | 84.13 | 78.87 | 6.20 | 77.67 |
| vt | 28.24 | 2.32 | 0.20 | 6.00 |

