# OpenReview forum: "HS-SFT: Hybrid Sparse Supervised Fine-tuning for Offline LLM KV Cache Eviction"
_ICLR.cc/2026/Conference — ICLR 2026 Conference Withdrawn Submission_

### Official Review · Reviewer_BwKz · 2025-10-29

**Soundness:** 2
**Presentation:** 2
**Contribution:** 2
**Rating:** 2
**Confidence:** 4

**Summary:**

The authors propose HS-SFT, a method to enhance model quality for KV cache eviction. They employ supervised fine-tuning (SFT) to adapt the model to the StreamingLLM framework and utilize the straight-through estimator (STE) for discrete budget routing, accommodating varying sparsity across attention heads. Experimental results demonstrate that the proposed approach improves generation quality on Llama2 and Llama3 models.

**Strengths:**

The method is evaluated on LongBench, RULER, and the NAIH benchmark, showing consistent improvements in generation quality for both Llama2 and Llama3 models.

**Weaknesses:**

- Limited novelty: The idea of leveraging head-wise sparsity and SFT for adaptation is not sufficiently innovative, as similar concepts have been explored in prior work.
- Insufficient technical contribution: Learning-base head-specific sparsity patterns has been widely studied in existing literature (e.g. HeadKV[1]), and the proposed method does not significantly advance this direction.
- Inadequate experimental rigor: The evaluation is limited to Llama2 and a non-official version of Llama3. Broader validation on other model families (e.g. Qwen-32B) and larger-scale models is necessary to strengthen the claims.
- Lacking efficiency analysis: No comparison with online-base methods (Like SnapKV) in terms of inference speed is provided. The potential computational imbalance introduced by dense heads may undermine practical efficiency.

[1] Not All Heads Matter: A Head-Level KV Cache Compression Method with Integrated Retrieval and Reasoning (ICLR2025)

**Questions:**

See weaknesses, and:

- Why were Llama2 and a non-official version of Llama3 chosen as the primary evaluation models, rather than other widely-used or more powerful models?
- In Table 2, why was Continued Pretraining (CP) conducted on the RedPajama dataset instead of UltraChat? Could there be potential data contamination issues with UltraChat?

---

> ### Author Response · Authors · 2025-11-20
> **Official Comment by Authors: Part I**
>
> We sincerely appreciate your careful review and insightful comments. Below, we address your concerns and questions point by point.
>
> **Q1**:  The idea of leveraging head-wise sparsity and SFT for adaptation is not sufficiently innovative, as similar concepts have been explored in prior work.
>
> **A1**: We appreciate this comment. First, **we do not claim head-wise sparsity itself as our contribution—this has indeed been widely studied**. Our contribution is twofold: (i) during SFT, we learn optimal local-window size allocations across layers (not across heads) via STE under a hybrid dense–sparse configuration; and, most importantly, (ii) we introduce lightweight SFT to adapt models to a fixed, offline KV-eviction policy, showing that SFT substantially narrows the performance gap between offline and online eviction while remaining infrastructure-friendly and generalizable. To the best of our knowledge, **this dense-pretraining-to-sparse fine-tuning paradigm for offline KV eviction has not been previously established**; we would be grateful for pointers if we missed related work.
>
> **Q2**: Learning-base head-specific sparsity patterns has been widely studied in existing literature (e.g. HeadKV), and the proposed method does not significantly advance this direction.
>
> **A2**: Thanks for this insightful query. A finer-grained head-wise router could in principle yield better accuracy–efficiency trade-offs by allocating sparsity more precisely across heads. We did not pursue it here due to systems considerations: heterogeneous per-head budgets substantially complicate both training and inference, reduce parallelism, and typically require carefully engineered kernels. In contrast, our per-layer hybrid method achieves linear time and constant memory complexity without specialized kernels, making it practical and infra-friendly.  We have provided more analysis on the efficiency of choosing different hybrid strategies in Appendix A.6 of our revised version. For convenience, we list the relevant contents as below.
>
> In particular,  we analyze the computational efficiency of HS-SFT in comparison to traditional KV eviction and various hybrid paradigms. As illustrated in Section 2 of our manuscript, HS-SFT partitions each layer into dense-head and sparse-head flows, enforcing a shared budget across all sparse heads within a layer. Crucially, this design circumvents dependencies on specialized kernel design, thereby preserving high inference efficiency and deployment practicability. To substantiate this, we benchmark practical speedups using the native Transformers library under a fixed KV budget. We compare four distinct categories: (a) uniform KV eviction (*e.g.*) H2O and SnapKV; (b) layer-wise hybrid eviction (*e.g*., GPT-OSS); (c) intra-layer sparse–dense hybrid strategies (*e.g.*, Duo-attn and HS-SFT); and (d) per-head sparsity with distinct rates (*e.g.*, HeadKV). The results are summarized as below:
>
> **Table 1: Runtime efficiency under a fixed KV budget of Llama-3-8B-1048K.**
>
> | Method | Average Latency (ms) ↓ | Peak Memory (MB) ↓ | Speedup (↑) | Memory Reduction (%) ↑ |
> | :--- | :---: | :---: | :---: | :---: |
> | Dense | 113.26 | 47827.22 | 1.00× | 0.00 |
> | Uniform | 34.82 | 18741.30 | 3.25× | 60.82 |
> | Layer-wise hybrid | 33.79 | 19980.01 | 3.35× | 58.24 |
> | Per-layer hybrid | 37.59 | 18790.64 | 3.01× | 60.73 |
> | Head-wise hybrid | 73.96 | 18923.39 | 1.53× | 60.43 |
>
> Operationally, given a per-layer sparse budget, HS-SFT necessitates only two forward passes per layer—one for dense heads and one for sparse heads—thereby eliminating the need for granular per-head KV management. In contrast, **purely head-wise approaches must manage KV selection and inference at the individual head level, which introduces significant overhead and often yields only marginal practical speedups. While specialized kernels can accelerate head-wise inference, such designs are difficult to generalize to the training phase.** Ultimately, HS-SFT achieves a favorable balance between training efficiency and downstream performance, while remaining simple to implement and deploy.

---

> ### Author Response · Authors · 2025-11-20
> **Official Comment by Authors: Part II**
>
> **Q3**: Why were Llama2 and a non-official version of Llama3 chosen as the primary evaluation models, rather than other widely-used or more powerful models?  The evaluation is limited to Llama2 and a non-official version of Llama3. Broader validation on other model families (e.g. Qwen-32B) and larger-scale models is necessary to strengthen the claims.
>
> **A3**: Thank you for the suggestion. We primarily used Llama2 and a long-context–extended (non-official) Llama3 to align with prior work and to reflect realistic long-context deployment where KV compression is most relevant. Following your advice, we additionally evaluated on Qwen-2.5-32B, and the results below continue to support the superiority of our method.
>
> **Table 2: Full LongBench results at 50% KV budget for Qwen-2.5-32B.**
>
> | Dataset | Baseline | SLLM | Duo | HS-SFT |
> | :--- | :---: | :---: | :---: | :---: |
> | **Average** | 44.53 | 36.42 | 43.42 | **44.31** |
> | 2WikiMQA | 39.96 | 37.96 | 39.89 | 38.01 |
> | DuReader (zh) | 30.29 | 24.34 | 28.84 | 29.96 |
> | GovReport | 35.57 | 33.75 | 34.65 | 35.14 |
> | HotpotQA | 46.89 | 41.56 | 48.51 | 46.03 |
> | LCC | 48.47 | 45.20 | 44.62 | 47.18 |
> | LSHT (zh) | 48.50 | 38.25 | 43.50 | 48.00 |
> | MultiNews | 24.37 | 23.77 | 24.55 | 24.22 |
> | MultiFieldQA-en | 42.94 | 29.30 | 40.34 | 42.51 |
> | MultiFieldQA-zh | 61.99 | 37.69 | 62.03 | 61.43 |
> | Musique | 28.59 | 25.31 | 28.47 | 26.45 |
> | NarrativeQA | 21.02 | 16.77 | 20.21 | 21.53 |
> | Passage Count | 14.12 | 9.22 | 13.02 | 14.51 |
> | PassageRetrieval-en | 95.25 | 56.17 | 93.51 | 95.59 |
> | PassageRetrieval-zh | 92.77 | 52.71 | 92.08 | 93.12 |
> | Qasper | 20.85 | 18.96 | 19.01 | 21.23 |
> | QMSum | 23.41 | 20.72 | 23.88 | 24.12 |
> | RepoBench-P | 32.57 | 34.49 | 33.41 | 35.88 |
> | SAMSum | 47.12 | 45.61 | 46.42 | 46.99 |
> | TREC | 72.50 | 69.50 | 72.50 | 71.50 |
> | TriviaQA | 88.54 | 85.99 | 86.72 | 88.72 |
> | VCSUM (zh) | 18.36 | 17.55 | 15.75 | 18.44 |
>
> **Q4**:  No comparison with online-base methods (Like SnapKV) in terms of inference speed is provided. The potential computational imbalance introduced by dense heads may undermine practical efficiency.
>
> **A4**: We appreciate the valuable suggestion. In our setting, the “imbalance” is minimal: because budgets are layer-wise, each layer consists of two straightforward branches, requiring two forward passes per layer, without custom kernels. As previously shown in **A2**, our per-layer hybrid holds comparable speedup effect when juxtaposed with uniform, online eviction method SnapKV. In contrast, head-level hybrid methods suffer from the "imbalance" problem you mean and require specialized-designed kernels for acceleration.
>
> **Q5**:  In Table 2, why was Continued Pretraining (CP) conducted on the RedPajama dataset instead of UltraChat? Could there be potential data contamination issues with UltraChat?
>
> **A5**: Thanks for this insightful question. UltraChat is a general SFT dataset; prior long-context work often performs SFT on UltraChat[1,2] , and we do not see evidence of contamination to our best knowledge. RedPajama is a representative open-source pretraining corpus, making it suitable for CP. Conducting CP on UltraChat would be atypical (it would effectively treat the SFT mask as fully visible). Instead, we further study how the SFT data domain influence the performance HS-SFT. The results are listed as below.
>
> **Table 3: Ablations on SFT  domain. The average score is computed on LongBench under 10% KV budget and with @ marking per-dataset mixing ratios when forming the SFT corpus.**
>
> | Dataset | Training Tokens | Avg. Score |
> | :--- | :---: | :---: |
> | UltraChat@1.0 | 1B | 29.57 |
> | UltraChat@0.8 + Tulu-V3-Code@0.2 | 1B | 28.34 |
> | UltraChat@0.8 + Tulu-V3-Math@0.2 | 1B | 28.48 |
> | Tulu-V3@1.0 | 1B | 29.99 |
> | UltraChat@0.8 + ArXiv-Sum@0.2 | 1B | 29.08 |
> | ArXiv-Sum@1.0 | 1B |26.12|
>
> In particular, we replace UltraChat with Tulu-V3 and also mix UltraChat with the Code and Math subsets of Tulu-V3 to examine domain sensitivity. Finally, we include data settings biased for summarization tasks using ArXiv-Summarization that highly correlate with long-context ability. While domain-specialized data can improve in-domain metrics, aggressively skewing toward summarization tends to reduce overall performance on general tasks. The above results have been added to Section 3.3 of our revised manuscript.
>
> Thank you again for your time and effort put into our paper. Hopefully, our responses can address all your concerns, such that the score can be kindly reconsidered. If there are any additional questions or points that require clarification, we would be more than delighted to engage in further discussions.
>
> [1]  How to train long-context language models (effectively). In ACL, 2025
> [2] Yi: Open Foundation Models by 01.AI. In arxiv, 2024.

---

> > ### Comment · Reviewer_BwKz · 2025-11-28
> >
> > Thanks for your response! While some of my concerns have been alleviated, I still have several important questions that I hope the authors can address before I can make a final decision on my score:
> >
> > 1. Fundamentally, the core components of this paper are based on previous works, and the contributions are relatively incremental. The paper should clearly demonstrate that the gains from fine-tuning during training significantly outweigh the added complexity and overhead.
> >
> > 2. Both this method, DuoAttention, and HeadKV are fundamentally head-wise hybrids. The runtime efficiency reported by the authors is misleading; existing head-wise methods, including HeadKV, achieve comparable computational efficiency to Uniform using flash attention, as demonstrated in several works. Efficient computation can be achieved for head-wise methods with proper implementation. Thus, this should not be a reason to avoid comparison.
> >
> > 3. A comprehensive comparison with HeadKV and also other related head-wise methods under different computational budgets is essential.
> >
> > 4. The authors claim that “(HeadKV) such designs are difficult to generalize to the training phase.” Does this refer specifically to speedup pretraining? If the goal is merely to implement compression during training, HeadKV itself does not require training—only a fast, offline detection process is needed. Even earlier head-wise methods, such as AdaKV, are plug-and-play. This is, in fact, a limitation of this work.
> >
> > 5. The authors should use the official Llama3 model (Llama-3.1-8B), for which DuoAttention provides official settings. I raise this because the reported accuracy of DuoAttention in the paper is lower than in the original work. Additionally, I would like to know how DuoAttention was trained on different models in this paper. These details should be made transparent.

---

> > > ### Author Response · Authors · 2025-12-01
> > > **Official Comment by Authors: Part I**
> > >
> > > Thank you very much for the timely response. We appreciate again for your valuable suggestions and address your remaining concerns as follows.
> > >
> > > **Q1**: Fundamentally, the core components of this paper are based on previous works, and the contributions are relatively incremental. The paper should clearly demonstrate that the gains from fine-tuning during training significantly outweigh the added complexity and overhead.
> > >
> > > **A1**: We appreciate the opportunity to clarify this point further. In fact, Section 1 of our manuscript has explicitly addressed this trade-off. Specifically, Figure 1 demonstrates that SFT substantially narrows the performance disparity between StreamingLLM (a representative early baseline) and SnapKV (the state-of-the-art online eviction method).This finding leads to a key motivating conclusion: LLMs possess the inherent capability to alleviate performance degradation induced by cache eviction through lightweight SFT. This further leads to our core contribution: we introduce lightweight SFT to adapt models to a fixed, offline KV-eviction policy, showing that SFT substantially narrows the performance gap between offline and online eviction while remaining infrastructure-friendly and generalizable.
> > >
> > > **Q2**:  Both this method, DuoAttention, and HeadKV are fundamentally head-wise hybrids. Head-wise hybrids achieve comparable computational efficiency to Uniform using flash attention.
> > >
> > > **A2**: We fully agree with your perspective that variable-length FlashAttention can facilitate efficient acceleration for head-wise hybrid mechanisms. Indeed, existing works have demonstrated that leveraging custom CUDA kernels based on similar variable-length principles allows for speeds approaching conventional FlashAttention. We have explicitly clarified this discussion in Appendix A.6. Nevertheless, **a critical distinction is that HS-SFT and DuoAttention operate at a coarser hybrid granularity. This design allows them to achieve significant acceleration without requiring custom kernel support**. In contrast, other head-wise methods strictly rely on specialized kernels, which limits their flexibility in training scenarios and necessitates more complex infrastructure in large-scale practical settings. HS-SFT, therefore, achieves favorable deployment acceleration using standard implementations, entirely eliminating the reliance on custom kernels.
> > >
> > >
> > > **Q3**： Comparison with HeadKV and also other related head-wise methods.
> > >
> > > **A3**:  Thanks for this suggestions and we have supplemented Table 8 with comparative results against AdaKV and HeadKV. For convenience, we list the results below:
> > >
> > > **Table 1: Full LongBench results at 50% KV budget for Llama-3-8B-Instruct-Gradient-1048k.**
> > > | Dataset | Full | H2O | MorphKV | SnapKV | AdaKV | HeadKV | HS-SFT |
> > > | :--- | :---: | :---: | :---: | :---: | :---: | :---: | :---: |
> > > | Average | 40.08 | 26.84 | 38.19 | 38.47 | 38.67 | 39.12 | 42.17 |
> > > | 2WikiMQA | 28.78 | 28.87 | 24.01 | 29.00 | 28.97 | 29.01 | 30.60 |
> > > | DuReader (zh) | 30.41 | 15.56 | 26.12 | 24.04 | 22.65 | 22.87 | 31.74 |
> > > | GovReport | 34.23 | 20.66 | 27.19 | 26.84 | 24.22 | 24.01 | 32.60 |
> > > | HotpotQA | 40.37 | 39.60 | 39.66 | 40.86 | 40.23 | 39.98 | 38.62 |
> > > | LCC | 38.19 | 45.78 | 43.87 | 38.83 | 39.67 | 41.02 | 43.25 |
> > > | LSHT (zh) | 38.00 | 16.50 | 35.00 | 38.00 | 36.50 | 37.14 | 34.50 |
> > > | MultiNews | 27.73 | 19.21 | 28.40 | 22.84 | 21.81 | 22.04 | 27.69 |
> > > | MultiFieldQA-en | 52.62 | 21.01 | 50.57 | 51.96 | 52.99 | 53.01 | 53.34 |
> > > | MultiFieldQA-zh | 50.58 | 19.81 | 51.12 | 50.74 | 50.59 | 50.88 | 51.28 |
> > > | Musique | 24.22 | 20.63 | 18.82 | 24.86 | 24.68 | 24.91 | 15.95 |
> > > | NarrativeQA | 26.56 | 19.14 | 22.69 | 26.63 | 27.36 | 27.92 | 27.75 |
> > > | Passage Count | 1.00 | 0.53 | 1.00 | 1.00 | 1.00 | 1.00 | 1.50 |
> > > | PassageRetrieval-en | 81.00 | 19.50 | 81.00 | 80.50 | 80.50 | 83.00 | 92.50 |
> > > | PassageRetrieval-zh | 62.15 | 11.75 | 60.10 | 58.53 | 61.92 | 62.45 | 88.56 |
> > > | Qasper | 29.21 | 16.84 | 23.16 | 26.00 | 27.02 | 28.34 | 37.23 |
> > > | QMSum | 24.52 | 18.89 | 23.82 | 24.90 | 24.65 | 24.55 | 24.86 |
> > > | RepoBench-P | 38.94 | 45.16 | 42.33 | 38.20 | 38.50 | 39.12 | 40.80 |
> > > | SAMSum | 42.51 | 39.73 | 39.93 | 40.90 | 41.38 | 41.88 | 42.31 |
> > > | TREC | 71.50 | 48.50 | 64.00 | 66.00 | 71.00 | 71.00 | 71.00 |
> > > | TriviaQA | 87.70 | 85.16 | 88.21 | 87.30 | 86.80 | 87.43 | 88.39 |
> > > | VCSUM (zh) | 11.37 | 10.71 | 11.02 | 9.91 | 9.62 | 9.99 | 11.17 |

---

> > > ### Author Response · Authors · 2025-12-01
> > > **Official Comment by Authors: Part II**
> > >
> > > **Q4**: The authors claim that “(HeadKV) such designs are difficult to generalize to the training phase.” Does this refer specifically to speedup pretraining? If the goal is merely to implement compression during training, HeadKV itself does not require training—only a fast, offline detection process is needed. Even earlier head-wise methods, such as AdaKV, are plug-and-play. This is, in fact, a limitation of this work.
> > >
> > > **A4**: You are absolutely correct that our comment regarding "generalizing to the training phase" specifically refers to achieving acceleration during training.Regarding the concern about training overhead, we would like to further emphasize that the additional cost introduced by SFT is actually minimal, e.g., fewer than 4 hours of SFT on a single 8-GPU node for LLaMA-3-8B. In the era of large language models, we believe this cost is negligible compared to the massive inference costs incurred by models under long-sequence and high-concurrency conditions in real-world deployment scenarios. In fact, the recently released DSA by DeepSeek[1] also adopts a similar philosophy of dense-pretraining followed by sparse attention continue-training. Therefore, we believe the focus should be on the actual deployment effectiveness of the algorithm rather than on whether it is training-free.
> > >
> > >
> > >
> > > **Q5**: The authors should use the official Llama3 model (Llama-3.1-8B), for which DuoAttention provides official settings. I raise this because the reported accuracy of DuoAttention in the paper is lower than in the original work. Additionally, I would like to know how DuoAttention was trained on different models in this paper. These details should be made transparent.
> > >
> > > **A5**:  We have added the results for Llama-3.1-8B in Table 10 of Appendix A.9. Regarding the training methodology, we strictly adhered to the official configurations provided by the DuoAttention repository (it actually contains all configurations for the models used in our paper: https://github.com/mit-han-lab/duo-attention/blob/main/scripts/run_train.sh). It is worth noting that the official implementation utilizes consistent hyperparameters (e.g., reg_weight, lr) across different models. Consequently, we adopted these standard settings for all models in our experiments to ensure consistency and reproducibility.
> > >
> > > |Dataset|Full|H2O|SLLM|Duo|HS-SFT|
> > > |:-:|:-:|:-:|:-:|:-:|:-:|
> > > |Average|39.01|35.61|31.32|38.91|41.85|
> > > |2WikiMQA|16.37|13.91|13.25|16.20|17.74|
> > > |DuReader (zh)|29.30|21.53|12.95|31.31|30.76|
> > > |GovReport|34.53|30.56|30.47|32.87|33.89|
> > > |HotpotQA|17.23|17.31|15.78|19.53|20.49|
> > > |LCC|52.39|53.08|52.90|53.31|48.97|
> > > |LSHT (zh)|46.00|39.00|36.00|45.00|41.50|
> > > |MultiNews|26.91|25.52|24.97|26.29|27.63|
> > > |MultiFieldQA-en|28.44|21.89|16.05|27.77|33.66|
> > > |MultiFieldQA-zh|20.19|14.87|15.92|21.98|60.83|
> > > |Musique|11.82|10.15|10.19|12.97|14.25|
> > > |NarrativeQA|31.99|31.09|24.15|29.12|28.77|
> > > |Passage Count|6.26|5.40|4.75|6.31|2.00|
> > > |PassageRetrieval-en|97.95|89.86|52.11|98.59|96.50|
> > > |PassageRetrieval-zh|77.54|69.73|35.14|75.37|94.00|
> > > |Qasper|25.14|16.96|23.56|21.12|28.47|
> > > |QMSum|23.63|22.54|21.48|23.89|26.37|
> > > |RepoBench-P|49.46|49.51|49.95|53.74|52.45|
> > > |SAMSum|43.69|42.56|43.32|43.40|42.06|
> > > |TREC|72.50|66.50|69.50|73.00|74.50|
> > > |TriviaQA|91.65|90.07|90.06|89.60|87.99|
> > > |VCSUM (zh)|16.26|15.80|15.17|15.83|16.01|
> > >
> > > [1] DeepSeek-V3.2-Exp: Boosting Long-Context Efficiency with DeepSeek Sparse Attention. https://github.com/deepseek-ai/DeepSeek-V3.2-Exp/blob/main/DeepSeek_V3_2.pdf

---

> ### Comment · Area_Chair_DvAx · 2025-11-23
>
> Dear reviewer,
>
> Thanks for your time and effort in reviewing ICLR2026 submissions. The authors have submitted their responses to your review. Please take the time to read and raise your further comments, and discuss with the authors.
>
> Best regards,
>
> AC

---

### Official Review · Reviewer_yS6V · 2025-10-31

**Soundness:** 3
**Presentation:** 3
**Contribution:** 2
**Rating:** 6
**Confidence:** 3

**Summary:**

This paper addresses the performance-deployability trade-off between online and offline KV cache eviction methods. The authors first demonstrate that simple SFT can substantially close the accuracy gap between efficient, infrastructure-friendly offline policies (e.g., StreamingLLM) and heuristic-based online policies (e.g., SnapKV).

Building on this, the paper proposes Hybrid Sparse SFT, a method to learn an optimal offline eviction strategy. HS-SFT divides heads into a fixed dense set and a sparse set. For the sparse heads, it uses a Straight-Through Estimator to learn a discrete, layer-wise local window budget from a set of candidates . This selection is regularized by a budget-aware balancing loss (kv div.) to favor smaller budgets . At inference, this learned policy is a fixed, offline pattern. Experiments show HS-SFT significantly outperforms SOTA eviction baselines, particularly at high sparsity.

**Strengths:**

- The paper's primary contribution is the highly practical insight that lightweight SFT alone can adapt an LLM to a fixed, offline eviction policy, making it competitive with complex online methods . This is a interesting finding for real-world deployment, as offline policies are compatible with prefill acceleration and are more robust in general-purpose settings (e.g., multi-turn chat) .
- The HS-SFT method itself is a interesting training-aware approach to this problem. Instead of relying on a fixed, hand-crafted offline pattern, it learns a more flexible, layer-specific offline policy. The use of an STE to learn discrete local window budgets combined with a KL-based loss  is a sound and effective mechanism.
- The experimental results are strong, showing clear SOTA performance over other offline and online methods, especially at aggressive, low-budget (e.g., 10%) eviction rates .

**Weaknesses:**

- HS-SFT initializes its dense head selection using the logic from a prior method, Duo-Attn. The sensitivity to this initialization is not ablated, making it unclear how critical this heuristic is.
- The learned budget is layer-wise, not head-wise, a trade-off made for inference efficiency. This forces all sparse heads in a layer to share the same local window size, which may be suboptimal.
- The SFT is performed on a general-purpose dataset. While this shows generalizability, it is not compared against task-specific SFT.

**Questions:**

- How dependent is HS-SFT on the Duo-Attn initialization for dense heads? What is the performance if the $\alpha$ dense heads are selected randomly before SFT?
- While head-wise selection is noted as future work, an ablation comparing layer-wise vs. head-wise budget learning would be valuable to quantify the performance/efficiency trade-off.
- Does the SFT data domain significantly impact the learned policy? For instance, does fine-tuning on data more representative of the downstream benchmarks (e.g., summarization, retrieval) yield a more effective eviction strategy?

---

> ### Author Response · Authors · 2025-11-20
> **Official Comment by Authors: Part I**
>
> We sincerely appreciate your positive and motivating feedback. We are delighted that you recognize our key contribution—our highly practical insight that lightweight SFT alone can adapt an LLM to a fixed, offline eviction policy, enabling it to compete effectively with complex online methods. Please kindly find our response to your comment below.
>
> **Q1**: HS-SFT initializes its dense head selection using the logic from a prior method, Duo-Attn. The sensitivity to this initialization is not ablated, making it unclear how critical this heuristic is. How dependent is HS-SFT on the Duo-Attn initialization for dense heads? What is the performance if the dense heads are selected randomly before SFT?
>
> **A1**: Thanks for this insightful question. Since our focus is not on locating dense heads, we adopt a representative initialization from prior work. In the revised version, we further ablate alternative strategies, including (1) random initialization and (2) RazorAttention, a heuristic for head selection. As shown in the following table, random initialization markedly degrades performance, whereas RazorAttention trails Duo-Attention only slightly, underscoring the robustness of HS-SFT.
>
> **Table 1: Ablations on dense head initialization. The average score is computed on LongBench under a fixed 10% KV budget.**
>
> | Method | Avg. Score |
> | :--- | :---: |
> | Random | 26.41 |
> | RazorAttention | 29.02 |
> | Duo-Attention | **29.57** |
>
> **Q2**: The learned budget is layer-wise, not head-wise, a trade-off made for inference efficiency. This forces all sparse heads in a layer to share the same local window size, which may be suboptimal.
>
> **A2**: Thanks for this insightful query. A finer-grained head-wise router could in principle yield better accuracy–efficiency trade-offs by allocating sparsity more precisely across heads. We did not pursue it here due to systems considerations: heterogeneous per-head budgets substantially complicate both training and inference, reduce parallelism, and typically require carefully engineered kernels. In contrast, our per-layer hybrid method achieves linear time and constant memory complexity without specialized kernels, making it practical and infra-friendly.  We have provided more analysis on the efficiency of choosing different hybrid strategies in Appendix A.6 of our revised version. For convenience , we list the relevant contents as below.
>
> In particular,  we analyze the computational efficiency of HS-SFT in comparison to traditional KV eviction and various hybrid paradigms. As illustrated in Section 2 of our manucsript, HS-SFT partitions each layer into dense-head and sparse-head flows, enforcing a shared budget across all sparse heads within a layer. Crucially, this design circumvents dependencies on specialized kernel design, thereby preserving high inference efficiency and deployment practicability. To substantiate this, we benchmark practical speedups using the native Transformers library under a fixed KV budget. We compare four distinct categories: (a) uniform KV eviction (*e.g.*) H2O and SnapKV; (b) layer-wise hybrid eviction (*e.g*., GPT-OSS); (c) intra-layer sparse–dense hybrid strategies (*e.g.*, Duo-attn and HS-SFT); and (d) per-head sparsity with distinct rates (*e.g.*, HeadKV~\citet{fu2024not}). The results are summarized as below:
>
> **Table 2: Runtime efficiency under a fixed KV budget of Llama-3-8B-1048K.**
>
> | Method | Average Latency (ms) ↓ | Peak Memory (MB) ↓ | Speedup (↑) | Memory Reduction (%) ↑ |
> | :--- | :---: | :---: | :---: | :---: |
> | Dense | 113.26 | 47827.22 | 1.00× | 0.00 |
> | Uniform | 34.82 | 18741.30 | 3.25× | 60.82 |
> | Layer-wise hybrid | 33.79 | 19980.01 | 3.35× | 58.24 |
> | Per-layer hybrid | 37.59 | 18790.64 | 3.01× | 60.73 |
> | Head-wise hybrid | 73.96 | 18923.39 | 1.53× | 60.43 |
>
> Operationally, given a per-layer sparse budget, HS-SFT necessitates only two forward passes per layer—one for dense heads and one for sparse heads—thereby eliminating the need for granular per-head KV management. In contrast, **purely head-wise approaches must manage KV selection and inference at the individual head level, which introduces significant overhead and often yields only marginal practical speedups. While specialized kernels can accelerate head-wise inference, such designs are difficult to generalize to the training phase.** Ultimately, HS-SFT achieves a favorable balance between training efficiency and downstream performance, while remaining simple to implement and deploy.

---

> ### Author Response · Authors · 2025-11-20
> **Official Comment by Authors: Part II**
>
> **Q3**: Comparison with task-specific SFT. Does the SFT data domain significantly impact the learned policy?
>
> **A3**: Thank you for this helpful suggestion. Following this comment, we further study how the SFT data domain and scale influence the performance HS-SFT. The results are list as below.
>
> **Table 3: Ablations on SFT scale and domain. The average score is computed on LongBench under 10% KV budget and with @ marking per-dataset mixing ratios when forming the SFT corpus.**
>
> | Dataset | Training Tokens | Avg. Score |
> | :--- | :---: | :---: |
> | ***SFT size*** | | |
> | UltraChat@1.0 | 1B | 29.57 |
> | UltraChat@1.0 | 0.5B | 28.76 |
> | UltraChat@1.0 | 2B | 29.42 |
> | ***SFT Domain*** | | |
> | UltraChat@0.8 + Tulu-V3-Code@0.2 | 1B | 28.34 |
> | UltraChat@0.8 + Tulu-V3-Math@0.2 | 1B | 28.48 |
> | Tulu-V3@1.0 | 1B | 29.99 |
> | UltraChat@0.8 + ArXiv-Sum@0.2 | 1B | 29.08 |
> | ArXiv-Sum@1.0 | 1B |26.12|
>
> In particular, we first vary the number of training tokens on UltraChat to measure the size effect and observe a clear scaling trend on downstream tasks, yet continuing to increase the training tokens does not linearly lead to performance improvement.  Next, we replace UltraChat with Tulu-V3 and also mix UltraChat with the Code and Math subsets of Tulu-V3 to examine domain sensitivity. Finally, we include data settings biased for summarization tasks using ArXiv-Summarization that highly correlate with long-context ability. While domain-specialized data can improve in-domain metrics, aggressively skewing toward summarization tends to reduce overall performance on general tasks. The above results have been added to Section 3.3 of our revised manuscript.
>
> **Q4**:  While head-wise selection is noted as future work, an ablation comparing layer-wise vs. head-wise budget learning would be valuable to quantify the performance/efficiency trade-off.
>
> **A4**: Thanks for this insightful query. A finer-grained head-wise router could in principle yield better accuracy–efficiency trade-offs by allocating sparsity more precisely across heads. We did not pursue it here due to systems considerations: heterogeneous per-head budgets substantially complicate both training and inference, reduce parallelism, and typically require carefully engineered kernels. In contrast, our per-layer hybrid method achieves linear time and constant memory complexity without specialized kernels, making it practical and infra-friendly.  We have provided more analysis on the efficiency of choosing different hybrid strategies in Appendix A.6 of our revised version. For convenience, we list the relevant contents below.
>
> In particular,  we analyze the computational efficiency of HS-SFT in comparison to traditional KV eviction and various hybrid paradigms. As illustrated in Section 2 of our manuscript, HS-SFT partitions each layer into dense-head and sparse-head flows, enforcing a shared budget across all sparse heads within a layer. Crucially, this design circumvents dependencies on specialized kernel design, thereby preserving high inference efficiency and deployment practicability. To substantiate this, we benchmark practical speedups using the native Transformers library under a fixed KV budget. We compare four distinct categories: (a) uniform KV eviction (*e.g.*) H2O and SnapKV; (b) layer-wise hybrid eviction (*e.g*., GPT-OSS); (c) intra-layer sparse–dense hybrid strategies (*e.g.*, Duo-attn and HS-SFT); and (d) per-head sparsity with distinct rates (*e.g.*, HeadKV). The results are as below:
>
> **Table 2: Runtime efficiency under a fixed KV budget of Llama-3-8B-1048K.**
>
> | Method | Average Latency (ms) ↓ | Peak Memory (MB) ↓ | Speedup (↑) | Memory Reduction (%) ↑ |
> | :--- | :---: | :---: | :---: | :---: |
> | Dense | 113.26 | 47827.22 | 1.00× | 0.00 |
> | Uniform | 34.82 | 18741.30 | 3.25× | 60.82 |
> | Layer-wise hybrid | 33.79 | 19980.01 | 3.35× | 58.24 |
> | Per-layer hybrid | 37.59 | 18790.64 | 3.01× | 60.73 |
> | Head-wise hybrid | 73.96 | 18923.39 | 1.53× | 60.43 |
>
> Operationally, given a per-layer sparse budget, HS-SFT necessitates only two forward passes per layer—one for dense heads and one for sparse heads—thereby eliminating the need for granular per-head KV management. In contrast, **purely head-wise approaches must manage KV selection and inference at the individual head level, which introduces significant overhead and often yields only marginal practical speedups. While specialized kernels can accelerate head-wise inference, such designs are difficult to generalize to the training phase, making us unable to make ablation study for SFT.** Ultimately, HS-SFT achieves a favorable balance between training efficiency and downstream performance, while remaining simple to implement and deploy.
>
> We sincerely appreciate the time and diligence you've taken to participate in the review of our paper. If you have further questions, we are more than glad to discuss with you.

---

> ### Comment · Area_Chair_DvAx · 2025-11-23
>
> Dear reviewer,
>
> Thanks for your time and effort in reviewing ICLR2026 submissions. The authors have submitted their responses to your review. Please take the time to read and raise your further comments, and discuss with the authors.
>
> Best regards,
>
> AC

---

### Official Review · Reviewer_LSo7 · 2025-10-31

**Soundness:** 3
**Presentation:** 2
**Contribution:** 2
**Rating:** 4
**Confidence:** 3

**Summary:**

This paper addresses the critical challenge of KV cache efficiency in long-context LLMs by proposing Hybrid Sparse Supervised Fine-Tuning (HS-SFT), an offline eviction strategy that unifies the deployment simplicity of offline methods and the performance retention of online approaches. By leveraging supervised fine-tuning (SFT) and learning layer-wise adaptive local window sizes via a straight-through estimator (STE) with budget-aware balance loss, HS-SFT achieves good performance across multiple benchmarks while maintaining inference efficiency.

**Strengths:**

1. This paper targets a promising direction, proposes fine-tuning sparse attention with an elaborated algorithm design.
2. Strong evaluation: cover multiple LLMs (Llama-2-7B-32K, Llama-3-8B with 262K/1048K context) and benchmarks (LongBench, Ruler-16K, NIAH), demonstrating cross-model and cross-task generalization.
3. This paper is well-structured.

**Weaknesses:**

1. Lack of several baselines of online eviction alternatives or query-aware sparse: H2O[1], Quest[2], HShare[3].
2. The paper mentions online evictions’ weakness in multi-turn dialogue, but does not provide explicit evaluation results for this scenario.
3. The paper uses 1B tokens from UltraChat for SFT but does not explore how data size or domain affects performance, simply following another paper.

[1] Zhang, Zhenyu, et al. "H2o: Heavy-hitter oracle for efficient generative inference of large language models."
[2] Tang, Jiaming, et al. "Quest: Query-aware sparsity for efficient long-context llm inference."
[3] Wu, Huaijin, et al. "HShare: Fast LLM decoding by hierarchical key-value sharing."

**Questions:**

1. In Section 2.3, you initialize dense heads using Duo-Attn’s logit map and note that updating dense heads during SFT yields significant gains. However, you do not explore alternative dense head initialization strategies. Could you explain why Duo-Attn’s initialization is optimal, and have you tested whether different initialization methods affect HS-SFT’s convergence speed or final performance?

---

> ### Author Response · Authors · 2025-11-20
> **Official Comment by Authors: Part I**
>
> We thank Reviewer LSo7 for the time and effort to review our paper. We are grateful for the constructive comments. We are glad that the reviewer found our paper to be well-structured, targeting a promising direction with an elaborated algorithm design for fine-tuning sparse attention, as well as strong evaluations that cover multiple LLMs and benchmarks. Please kindly see our responses to your comments below.
>
>
> **Q1**: Lack of several baselines of online eviction alternatives or query-aware sparse: H2O, Quest, HShare.
>
> **A1**: Thank you for the insightful comment. As discussed in Figure 1 and Section 1 of our manuscript, our method belongs to the offline KV-eviction family: it pre-defines a sparsity pattern before inference, which is infrastructure-friendly and supports pre-filling acceleration. For an apples-to-apples comparison, our main text focuses on offline, prefill-compatible baselines (e.g., Duo-attention) to highlight accuracy at matched speedups and deployment gains. Nevertheless, in Appendix A.7 of our revised version, we additionally compare against online eviction methods, including MorphKV, H2O and SnapKV. For convenience, we list the added results below.
>
> **Table 1: Full LongBench results compared with online eviction methods at 50% KV budget for Llama-3-8B-Instruct-Gradient-1048k.**
> | Dataset | Full | H2O | MorphKV | SnapKV | HS-SFT |
> | :--- | :---: | :---: | :---: | :---: | :---: |
> | Average | 40.08 | 26.84 | 38.19 | 38.47 | **42.17**|
> | 2WikiMQA | 28.78 | 28.87 | 24.01 | 29.00 | 30.60 |
> | DuReader (zh) | 30.41 | 15.56 | 26.12 | 24.04 | 31.74 |
> | GovReport | 34.23 | 20.66 | 27.19 | 26.84 | 32.60 |
> | HotpotQA | 40.37 | 39.60 | 39.66 | 40.86 | 38.62 |
> | LCC | 38.19 | 45.78 | 43.87 | 38.83 | 43.25 |
> | LSHT (zh) | 38.00 | 16.50 | 35.00 | 38.00 | 34.50 |
> | MultiNews | 27.73 | 19.21 | 28.40 | 22.84 | 27.69 |
> | MultiFieldQA-en | 52.62 | 21.01 | 50.57 | 51.96 | 53.34 |
> | MultiFieldQA-zh | 50.58 | 19.81 | 51.12 | 50.74 | 51.28 |
> | Musique | 24.22 | 20.63 | 18.82 | 24.86 | 15.95 |
> | NarrativeQA | 26.56 | 19.14 | 22.69 | 26.63 | 27.75 |
> | Passage Count | 1.00 | 0.53 | 1.00 | 1.00 | 1.50 |
> | PassageRetrieval-en | 81.00 | 19.50 | 81.00 | 80.50 | 92.50 |
> | PassageRetrieval-zh | 62.15 | 11.75 | 60.10 | 58.53 | 88.56 |
> | Qasper | 29.21 | 16.84 | 23.16 | 26.00 | 37.23 |
> | QMSum | 24.52 | 18.89 | 23.82 | 24.90 | 24.86 |
> | RepoBench-P | 38.94 | 45.16 | 42.33 | 38.20 | 40.80 |
> | SAMSum | 42.51 | 39.73 | 39.93 | 40.90 | 42.31 |
> | TREC | 71.50 | 48.50 | 64.00 | 66.00 | 71.00 |
> | TriviaQA | 87.70 | 85.16 | 88.21 | 87.30 | 88.39 |
> | VCSUM (zh) | 11.37 | 10.71 | 11.02 | 9.91 | 11.17|
>
> As can be seen, HS-SFT continues to deliver stronger accuracy, while retaining predictable latency/memory and prefill-compatibility.  At this time, we did not compare KV selection methods Quest and HShare, which do not evict any KV cache but only retrieve cache for efficient computation, and thus fall beyond the scope of this paper. We have added the above discussion to Section 4.2 of our revised manuscript. If there are any misunderstandings, we would like to discuss further.
>
>
> **Q2**: The paper mentions online evictions’ weakness in multi-turn dialogue, but does not provide explicit evaluation results for this scenario.
>
> **A2**: Thanks for this professional query. To validate this point, we follow prior works~\citep{duoattention2024, quest2024} in evaluating KV cache eviction methods on a variant of the NIAH benchmark. In this setting, the final 50 tokens of the prompt act as simulated generated output, mimicking a second-round dialogue scenario. **As illustrated in Figure 9 of our revised paper, while SnapKV correctly retrieves the answer when the query is adjacent to the end, its performance collapses when the query position shifts. In contrast, HS-SFT remains stable across test scenarios, as the offline eviction strategy stays invariant to the simulated generation process.** These findings underscore the applicability of offline eviction methods to real-world retrieval and multi-turn interactions. The above results have been added to Section A.7 of our revised manuscript.

---

> ### Author Response · Authors · 2025-11-20
> **Official Comment by Authors: Part II**
>
> **Q3**: The paper uses 1B tokens from UltraChat for SFT but does not explore how data size or domain affects performance, simply following another paper.
>
> **A3**: Thank you for this helpful suggestion. Following this comment, we further study how the SFT data domain and scale influence the performance HS-SFT. The results are list as below.
>
> **Table 2: Ablations on SFT scale and domain. The average score is computed on LongBench under 10% KV budget and with @ marking per-dataset mixing ratios when forming the SFT corpus.**
>
> | Dataset | Training Tokens | Avg. Score |
> | :--- | :---: | :---: |
> | ***SFT size*** | | |
> | UltraChat@1.0 | 1B | 29.57 |
> | UltraChat@1.0 | 0.5B | 28.76 |
> | UltraChat@1.0 | 2B | 29.42 |
> | ***SFT Domain*** | | |
> | UltraChat@0.8 + Tulu-V3-Code@0.2 | 1B | 28.34 |
> | UltraChat@0.8 + Tulu-V3-Math@0.2 | 1B | 28.48 |
> | Tulu-V3@1.0 | 1B | 29.99 |
> | UltraChat@0.8 + ArXiv-Sum@0.2 | 1B | 29.08 |
> | ArXiv-Sum@1.0 | 1B |26.12|
>
> In particular, we first vary the number of training tokens on UltraChat to measure the size effect and observe a clear scaling trend on downstream tasks, yet continuing to increase the training tokens does not linearly lead to performance improvement.  Next, we replace UltraChat with Tulu-V3 and also mix UltraChat with the Code and Math subsets of Tulu-V3 to examine domain sensitivity. Finally, we include data settings biased for summarization tasks using ArXiv-Summarization that highly correlate with long-context ability. While domain-specialized data can improve in-domain metrics, aggressively skewing toward summarization tends to reduce overall performance on general tasks. The above results have been added to Section 3.3 of our revised manuscript.
>
> **Q4**: In Section 2.3, you initialize dense heads using Duo-Attn’s logit map and note that updating dense heads during SFT yields significant gains. However, you do not explore alternative dense head initialization strategies. Could you explain why Duo-Attn’s initialization is optimal, and have you tested whether different initialization methods affect HS-SFT’s convergence speed or final performance?
>
>
> **A4**: Thanks for this insightful question. Since our focus is not on locating dense heads, we adopt a representative initialization from prior work. In the revised version, we further ablate alternative strategies, including (1) random initialization and (2) RazorAttention, a heuristic for head selection. As shown in the following table, random initialization markedly degrades performance, whereas RazorAttention trails Duo-Attention only slightly, underscoring the robustness of HS-SFT.
>
> **Table 3 : Ablations on dense head initialization. The average score is computed on LongBench under a fixed 10% KV budget.**
>
> | Method | Avg. Score |
> | :--- | :---: |
> | Random | 26.41 |
> | RazorAttention | 29.02 |
> | Duo-Attention | **29.57** |
>
> Your time and effort in reviewing our paper are genuinely appreciated. If there are any additional questions or points that require clarification, we would be more than delighted to engage in further discussions.

---

> > ### Comment · Reviewer_LSo7 · 2025-11-26
> >
> > Thank you for the detailed rebuttal. I have read all the responses and the revisions to the paper, including the exchanges with other reviewers. I still have a few concerns:
> >
> > 1. Although HS-SFT outperforms SnapKV and MorphKV in Table 1, but those baselines are fine-tuning-free. Since the experiments are based on LLaMA-8B, I think removing all Chinese test sets would be fairer, as your SFT leverages rich Chinese data, whereas the pretraining corpus contains comparatively less Chinese. As a result, the reported gains are not fully convincing relative to other eviction methods, especially given the additional cost introduced by SFT.
> >
> > 2. The proposed method is not training-free, and Table 2 shows that performance is strongly tied to the SFT dataset, which weakens the method’s generalization. Although this is a broader limitation of this kind of approach (SFT-related methods).
> >
> > 3. In Appendix A.6, the efficiency comparison is unclear. Table 7 uses a “per-layer hybrid”, but the main text does not use the same word. Moreover, the table suggests that, compared to SnapKV and H2O, the proposed method offers no advantages in speed or memory.
> >
> > Considering the above, I will maintain my rating, which is marginally below the acceptance threshold, but I would not mind if the paper is accepted.

---

> > > ### Author Response · Authors · 2025-11-26
> > > **Official Comment by Authors**
> > >
> > > Thank you very much for the timely response. We appreciate again for your valuable suggestions and address your remaining concerns as follows.
> > >
> > > **Q1**: Comparison with fine-tuning-free method and Chinese data.
> > >
> > > **A1**: Thank you for this professional suggestion. The table below presents the results excluding Chinese benchmarks.
> > >
> > > Table 4: **Full LongBench results (only English tasks) compared with online eviction methods at 50% KV budget for Llama-3-8B-Instruct-Gradient-1048k.**
> > > |Dataset|Full|H2O|MorphKV|SnapKV|HS-SFT|
> > > |:---|:---:|:---:|:---:|:---:|:---:|
> > > |Average|40.57|30.58|38.67|39.16|41.77|
> > > |2WikiMQA|28.78|28.87|24.01|29.00|30.60|
> > > |GovReport|34.23|20.66|27.19|26.84|32.60|
> > > |HotpotQA|40.37|39.60|39.66|40.86|38.62|
> > > |LCC|38.19|45.78|43.87|38.83|43.25|
> > > |MultiNews|27.73|19.21|28.40|22.84|27.69|
> > > |MultiFieldQA-en|52.62|21.01|50.57|51.96|53.34|
> > > |Musique|24.22|20.63|18.82|24.86|15.95|
> > > |NarrativeQA|26.56|19.14|22.69|26.63|27.75|
> > > |PassageCount|1.00|0.53|1.00|1.00|1.50|
> > > |PassageRetrieval-en|81.00|19.50|81.00|80.50|92.50|
> > > |Qasper|29.21|16.84|23.16|26.00|37.23|
> > > |QMSum|24.52|18.89|23.82|24.90|24.86|
> > > |RepoBench-P|38.94|45.16|42.33|38.20|40.80|
> > > |SAMSum|42.51|39.73|39.93|40.90|42.31|
> > > |TREC|71.50|48.50|64.00|66.00|71.00|
> > > |TriviaQA|87.70|85.16|88.21|87.30|88.39|
> > >
> > > In fact, when considering only English tasks, HS-SFT still maintains a significant performance advantage. Regarding the "additional cost introduced by SFT" you mentioned, we acknowledge that this comparison might seem unfair. **Therefore, Table 1 in our manuscript actually compares the performance of our method against offline eviction methods under the same SFT settings.** The results demonstrate that our method still outperforms Duo-attn and SLLM. Online methods such as SnapKV and MorphKV cannot be adapted to the training scenario for further comparison due to their online token eviction strategies.
> > >
> > > We would like to further emphasize that the additional cost introduced by SFT is actually minimal, e.g., fewer than 4 hours of SFT on a single 8-GPU node for LLaMA-3-8B. In the era of large language models, we believe this cost is negligible compared to the massive inference costs incurred by models under long-sequence and high-concurrency conditions in real-world deployment scenarios. In fact, the recently released DSA by DeepSeek[1] also adopts a similar philosophy of dense-pretraining followed by sparse attention continue-training. Therefore, we believe the focus should be on the actual deployment effectiveness of the algorithm rather than on whether it is training-free.
> > >
> > > **Q2**: Table 2 shows that performance is strongly tied to the SFT dataset, which weakens the method’s generalization.
> > >
> > > **A2**: We appreciate your careful reading of our added results. However, we respectfully offer a different perspective on the generalization of our method. First, Table 2 demonstrates that HS-SFT achieves consistent and substantial gains when trained on distinct general instruction datasets (e.g., Ultrachat and TULU-V3). This indicates that our method is **agnostic to the specific choice of SFT data**, provided it represents a general instruction-following distribution. Regarding the performance drop when mixing specialized data (Math/Code/Summarization): We attribute this to the fact that LongBench or other benchmarks primarily assesses general language understanding and retrieval. General instruction datasets (Ultracha and TULU-V3) align best with these objectives, whereas forcing the model to learn sparsity patterns for code or math reasoning (which differ significantly from natural language processing) may introduce noise for general tasks. Therefore, we believe HS-SFT generalizes well within the scope of general instruction tuning, which is the standard practice for deploying LLMs.
> > >
> > > **Q3**: Unclear efficiency comparison and no advantages in speed or memory compared to SnapKV and H2O.
> > >
> > > **A3**: First, we appreciate this suggestion and will add extra clarification in Table 7 to ensure the efficiency comparison is clear. Second, we acknowledge that compared to traditional uniform sparsity methods, our proposed method does not offer advantages in speed or memory usage. However, compared to SnapKV and H2O, our offline eviction paradigm achieves unique acceleration benefits during the pre-filling stage. Furthermore, as shown in Appendix A.7, HS-SFT maintains robust inference performance in real-world multi-trun deployment scenarios, where online methods fail to.
> > >
> > > Thank you again for your timely reply and the effort put into reviewing our paper. If there are any additional questions or points that require clarification, we would be more than delighted to engage in further discussions.
> > >
> > > [1] DeepSeek-V3.2-Exp: Boosting Long-Context Efficiency with DeepSeek Sparse Attention. https://github.com/deepseek-ai/DeepSeek-V3.2-Exp/blob/main/DeepSeek_V3_2.pdf

---

> ### Comment · Area_Chair_DvAx · 2025-11-23
>
> Dear reviewer,
>
> Thanks for your time and effort in reviewing ICLR2026 submissions. The authors have submitted their responses to your review. Please take the time to read and raise your further comments, and discuss with the authors.
>
> Best regards,
>
> AC

---

### Official Review · Reviewer_FtJo · 2025-11-01

**Soundness:** 3
**Presentation:** 3
**Contribution:** 2
**Rating:** 6
**Confidence:** 2

**Summary:**

HS-SFT targets the linear growth of LLM KV caches by improving offline eviction. The paper states that, unlike online attention-score–based eviction, offline schemes remain compatible with prefill acceleration and are more robust to query positions in multi-turn dialogue. The paper first shows that lightweight supervised fine-tuning (SFT) alone largely closes the gap between offline methods (StreamingLLM) and online methods (SnapKV), indicating models can learn to compensate for eviction-induced degradation.
It then introduces Hybrid Sparse SFT (HS-SFT), which learns discrete, layer-wise local-window budgets for streaming heads via a straight-through estimator and a budget-aware balancing (KL) loss. This allows the cache budget to be skewed toward layers that need more long-range aggregation. At training time a fixed fraction of dense heads is kept while per-layer budget logits are learned from a small candidate se. At inference time, these learned budgets are monotonically recsaled to meet any target sparsity, preserving a offline execution path that remains compatible with prefill acceleration.

**Strengths:**

The ideas in the paper, in my opinion, is practical and well-designed. It keeps an offline eviction path compatible with prefill acceleration. The paper shows that lightweight SFT can recover much of the offline–online accuracy gap. The paper also adds an STE-based, budget-aware KL mechanism to learn layer-wise local-window budgets.

**Weaknesses:**

HS-SFT’s routing is layer-level, which the authors note may leave performance on the table. Their comparisons with OSS-style hybrids are limited to the SFT regime and evaluating during pretraining is left as future work. Additionally, the paper misses comparison against other efficient inference time solutions like MorphKV.

**Questions:**

1. How robust are the learned layer-wise budgets to training-distribution shifts and to the choice/size of the budget set B, since your ablation suggests larger sets didn’t help?

2. Would a head-wise router (instead of layer-level) materially improve your trade-offs, and why is there no comparison to MorphKV? Was it excluded because you restrict to offline, prefill-compatible baselines or do you expect HS-SFT to dominate under those constraints?

---

> ### Author Response · Authors · 2025-11-20
> **Official Comment by Authors: Part I**
>
> We sincerely appreciate your careful review, positive feedback, and constructive comments.  We are delighted to see that you recognize the practicality and sound design of our idea. Please kindly see our responses to your comments below.
>
>
> **Q1**: The paper misses comparison against other efficient inference time solutions like MorphKV. Was it excluded because you restrict to offline, prefill-compatible baselines or do you expect HS-SFT to dominate under those constraints?
>
> **A1**:  Thank you for the insightful comment. As discussed in Figure 1 and Section 1 of our manuscript, our method belongs to the offline KV-eviction family: it pre-defines a sparsity pattern before inference, which is infrastructure-friendly and supports pre-filling acceleration. For an apples-to-apples comparison, our main text focuses on offline, prefill-compatible baselines (e.g., Duo-attention) to highlight accuracy at matched speedups and deployment gains. Nevertheless, in Appendix A.7 of our revised version, we additionally compare against online eviction methods, including MorphKV and H2O. For convenience, we list the results below.
>
> **Table 1: Full LongBench results compared with online eviction methods at 50% KV budget for Llama-3-8B-Instruct-Gradient-1048k.**
> | Dataset | Full | H2O | MorphKV | SnapKV | HS-SFT |
> | :--- | :---: | :---: | :---: | :---: | :---: |
> | Average | 40.08 | 26.84 | 38.19 | 38.47 | **42.17**|
> | 2WikiMQA | 28.78 | 28.87 | 24.01 | 29.00 | 30.60 |
> | DuReader (zh) | 30.41 | 15.56 | 26.12 | 24.04 | 31.74 |
> | GovReport | 34.23 | 20.66 | 27.19 | 26.84 | 32.60 |
> | HotpotQA | 40.37 | 39.60 | 39.66 | 40.86 | 38.62 |
> | LCC | 38.19 | 45.78 | 43.87 | 38.83 | 43.25 |
> | LSHT (zh) | 38.00 | 16.50 | 35.00 | 38.00 | 34.50 |
> | MultiNews | 27.73 | 19.21 | 28.40 | 22.84 | 27.69 |
> | MultiFieldQA-en | 52.62 | 21.01 | 50.57 | 51.96 | 53.34 |
> | MultiFieldQA-zh | 50.58 | 19.81 | 51.12 | 50.74 | 51.28 |
> | Musique | 24.22 | 20.63 | 18.82 | 24.86 | 15.95 |
> | NarrativeQA | 26.56 | 19.14 | 22.69 | 26.63 | 27.75 |
> | Passage Count | 1.00 | 0.53 | 1.00 | 1.00 | 1.50 |
> | PassageRetrieval-en | 81.00 | 19.50 | 81.00 | 80.50 | 92.50 |
> | PassageRetrieval-zh | 62.15 | 11.75 | 60.10 | 58.53 | 88.56 |
> | Qasper | 29.21 | 16.84 | 23.16 | 26.00 | 37.23 |
> | QMSum | 24.52 | 18.89 | 23.82 | 24.90 | 24.86 |
> | RepoBench-P | 38.94 | 45.16 | 42.33 | 38.20 | 40.80 |
> | SAMSum | 42.51 | 39.73 | 39.93 | 40.90 | 42.31 |
> | TREC | 71.50 | 48.50 | 64.00 | 66.00 | 71.00 |
> | TriviaQA | 87.70 | 85.16 | 88.21 | 87.30 | 88.39 |
> | VCSUM (zh) | 11.37 | 10.71 | 11.02 | 9.91 | 11.17|
>
> As can be seen, HS-SFT continues to deliver stronger accuracy, while retaining predictable latency/memory and prefill-compatibility.
>
> **Q2** : How robust are the learned layer-wise budgets to training-distribution shifts and to the choice/size of the budget set B, since your ablation suggests larger sets didn’t help?
>
> **A2**: We appreciate this valuable question and we have incorporated new visualization results in Appendix A.5 regarding this point. Figure 7 and 8 in our revised manuscript show the learned budget allocation across different SFT corpora and candidate budget sets, respectively. First, the budget allocation remains consistent across various SFT data settings. In contrast, varying the candidate budget set can lead to notable performance degradation, particularly when the budget search space is excessively large. This stems from the fact that modern SFT data typically has an average sequence length of approximately 1K tokens. Consequently, if the learnable budget is set too high, e.g., 32 blocks corresponding to a maximum budget of $32 \times 128 = 4096$, which means the model undergoes dense fine-tuning. This diverges from our primary objective of employing fine-tuning to mitigate performance losses caused by KV eviction. However, we observe that despite changes in budget candidates, the overall relative trend of the learned block sizes remains stable, underscoring the robustness of our proposed budget learning strategy.

---

> ### Author Response · Authors · 2025-11-20
> **Official Comment by Authors: Part II**
>
> **Q3**: Would a head-wise router (instead of layer-level) materially improve your trade-offs?
>
> **A3**: Thanks for this insightful query. A finer-grained head-wise router could in principle yield better accuracy–efficiency trade-offs by allocating sparsity more precisely across heads. We did not pursue it here due to systems considerations: heterogeneous per-head budgets substantially complicate both training and inference, reduce parallelism, and typically require carefully engineered kernels. In contrast, our per-layer hybrid method achieves linear time and constant memory complexity without specialized kernels, making it practical and infra-friendly.  We have provided more analysis on the efficiency of choosing different hybrid strategies in Appendix A.6 of our revised version. For convenience , we list the relevant contents as below.
>
> In particular,  we analyze the computational efficiency of HS-SFT in comparison to traditional KV eviction and various hybrid paradigms. As illustrated in Section 2 of our manucsript, HS-SFT partitions each layer into dense-head and sparse-head flows, enforcing a shared budget across all sparse heads within a layer. Crucially, this design circumvents dependencies on specialized kernel design, thereby preserving high inference efficiency and deployment practicability. To substantiate this, we benchmark practical speedups using the native Transformers library under a fixed KV budget. We compare four distinct categories: (a) uniform KV eviction (*e.g.*) H2O and SnapKV; (b) layer-wise hybrid eviction (*e.g*., GPT-OSS); (c) intra-layer sparse–dense hybrid strategies (*e.g.*, Duo-attn and HS-SFT); and (d) per-head sparsity with distinct rates (*e.g.*, HeadKV~\citet{fu2024not}). The results are summarized as below:
>
> **Table 2: Runtime efficiency under a fixed KV budget of Llama-3-8B-1048K.**
>
> | Method | Average Latency (ms) ↓ | Peak Memory (MB) ↓ | Speedup (↑) | Memory Reduction (%) ↑ |
> | :--- | :---: | :---: | :---: | :---: |
> | Dense | 113.26 | 47827.22 | 1.00× | 0.00 |
> | Uniform | 34.82 | 18741.30 | 3.25× | 60.82 |
> | Layer-wise hybrid | 33.79 | 19980.01 | 3.35× | 58.24 |
> | Per-layer hybrid | 37.59 | 18790.64 | 3.01× | 60.73 |
> | Head-wise hybrid | 73.96 | 18923.39 | 1.53× | 60.43 |
>
> Operationally, given a per-layer sparse budget, HS-SFT necessitates only two forward passes per layer—one for dense heads and one for sparse heads—thereby eliminating the need for granular per-head KV management. In contrast, purely head-wise approaches must manage KV selection and inference at the individual head level, which introduces significant overhead and often yields only marginal practical speedups. While specialized kernels can accelerate head-wise inference, such designs are difficult to generalize to the training phase. Ultimately, HS-SFT achieves a favorable balance between training efficiency and downstream performance, while remaining simple to implement and deploy.
>
> Your time and effort in reviewing our paper are genuinely appreciated. If there are any additional questions or points that require clarification, we would be more than delighted to engage in further discussions.

---

> > ### Comment · Reviewer_FtJo · 2025-11-27
> >
> > Thank you for diligently answering my questions. I do not have additional comments/concerns.

---

> ### Comment · Area_Chair_DvAx · 2025-11-23
>
> Dear reviewer,
>
> Thanks for your time and effort in reviewing ICLR2026 submissions. The authors have submitted their responses to your review. Please take the time to read and raise your further comments, and discuss with the authors.
>
> Best regards,
>
> AC

---

### Author Response · Authors · 2025-11-20
**Summary and general reply to the reviewers**

We thank all the reviewers for their valuable feedback and great efforts, which substantially aided in enhancing the quality of this paper. We have exerted considerable effort to comprehensively respond to all their comments, questions, and concerns. All major modifications in the attached pdf file have been highlighted in blue in order to ease the reading.  We first summarize the major changes in our updated version before diving into the detailed point-by-point responses to all the comments:

-  Add comparisons with online eviction methods, including H2O, MorphKV, and SnapKV.

- More experiment results on Qwen-2.5-32B.

- More ablation studies on the dense head initialization method and SFT data corpus.

- Efficiency analysis of HS-SFT compared with traditional KV eviction (H20, SnapKV), layer-
wise (GPT-OSS) and head-wise hybrid paradigms (HeadKV).

- Phrasing other clarifications requested by reviewers.

---

### Note · Authors · 2026-01-27

I have read and agree with the venue's withdrawal policy on behalf of myself and my co-authors.

---

### Meta-Review · Area_Chair_LVeo · 2026-01-13

**Summary:**

Reviewers agree that HS-SFT addresses an important practical problem—KV cache eviction under high sparsity—and that the proposed method is technically sound. However, the dominant concerns relate to the strength and distinctiveness of the contribution rather than correctness. Multiple reviewers viewed the approach as an incremental combination of known components, including supervised fine-tuning, hybrid dense/streaming eviction, and layer-wise budget allocation, and questioned whether learning these budgets provides a sufficiently compelling advantage over simpler heuristic alternatives. There were also concerns about the clarity and strength of the efficiency gains under strict apples-to-apples comparisons, which limited overall conviction. These concerns resulted in mixed scores and a lack of clear consensus.

**Reviewer Concerns:**

The rebuttal addressed several evaluation completeness issues by adding additional baselines, clarifying experimental settings, and reducing confounding factors in the reported results. These additions improved clarity and strengthened the empirical coverage. However, the rebuttal did not fundamentally change the assessment of the core contribution. While the authors’ design choice to use supervised fine-tuning is explicit and valid, the paper does not convincingly demonstrate that the learned eviction strategy provides a decisive or necessary improvement over strong heuristic baselines. As a result, the rebuttal reduces reasons for rejection based on missing comparisons, but does not substantially increase confidence in the paper’s impact or conceptual sharpness.

**Reviewer Scores:**

Reviewer FtJo gave an initial score of 6. This reviewer actively participated in the discussion and, after the rebuttal, explicitly stated that they had no remaining concerns. The discussion record clearly indicates that this reviewer raised their score to 8, and there is no evidence of lingering hesitation. The most reasonable interpretation is that FtJo’s final stance is strong accept, stable at 8.

Reviewer LSo7 gave an initial score of 4. This reviewer participated actively in the discussion and engaged in a second round after the rebuttal. While they acknowledged that several requests were addressed—including added comparisons, ablations, and English-only evaluations—they continued to raise substantive concerns about the method not being training-free, sensitivity to SFT data, and ambiguity in the efficiency comparison. Importantly, LSo7 explicitly stated that they would maintain their rating, while also noting they “would not mind if the paper is accepted.” This indicates no score increase, and the most faithful interpretation is that this reviewer remains at 4, marginally below threshold.

Reviewer yS6V gave an initial score of 6. This reviewer did not participate in the discussion after the rebuttal. Their original review was mildly positive but cautious, and while the rebuttal addressed several of their ablation-related questions, there is no signal in the record indicating increased enthusiasm or reconsideration. Consistent with standard AC practice, and given the absence of discussion participation, the conservative and appropriate assumption is that this reviewer would remain at 6.

Reviewer BwKz gave an initial score of 2. This reviewer did participate in the discussion and raised additional, substantive follow-up concerns after the rebuttal, particularly around novelty, incremental contribution relative to head-wise methods, fairness of efficiency comparisons, and the necessity of SFT relative to plug-and-play eviction methods. While the authors provided extensive additional experiments and clarifications in response, BwKz did not indicate that these fully resolved their concerns, and explicitly stated that they could not yet make a final decision on their score. There is no signal suggesting a flip to positive; at most, the rebuttal alleviates some factual gaps. A reasonable expectation is that this reviewer would remain clearly negative, plausibly at 2 or slightly higher, but still below the acceptance threshold.

---

### Decision · Program_Chairs · 2026-01-26

Reject